# BORDER proteins protect expression of neighboring genes by promoting 3′ Pol II pausing in plants

Xuhong Yu[1,3], Pascal G.P. Martin [1,2,3] & Scott D. Michaels [1]

Ensuring that one gene's transcription does not inappropriately affect the expression of its neighbors is a fundamental challenge to gene regulation in a genomic context. In plants, which lack homologs of animal insulator proteins, the mechanisms that prevent transcriptional interference are not well understood. Here we show that BORDER proteins are enriched in intergenic regions and prevent interference between closely spaced genes on the same strand by promoting the 3′ pausing of RNA polymerase II at the upstream gene. In the absence of BORDER proteins, 3′ pausing associated with the upstream gene is reduced and shifts into the promoter region of the downstream gene. This is consistent with a model in which BORDER proteins inhibit transcriptional interference by preventing RNA polymerase from intruding into the promoters of downstream genes.

[1] Department of Biology, Indiana University, 915 East Third Street, Bloomington, IN 47405, USA. [2] Toxalim (Research Centre in Food Toxicology), Université de Toulouse, INRA, ENVT, INP-Purpan, UPS, 31027 Toulouse, France. [3]These authors contributed equally: Xuhong Yu, Pascal G. P. Martin. Correspondence and requests for materials should be addressed to S.D.M. (email: michaels@indiana.edu)

Transcription of a gene does not occur in isolation but within the context of its genomic environment. The transcription of one gene has the potential to influence or interfere with that of its neighbors. Transcriptional interference (TI) can take many forms but is broadly defined as the direct negative impact of one gene's transcription on a second gene that is located in *cis*[1]. For example, if two genes are oriented in tandem on the same DNA strand, it is possible for the elongating RNA Polymerase II (Pol II) from the upstream gene to intrude into the promoter region of the downstream gene. This "promoter intrusion" has the potential to interfere with the binding of transcription factors, assembly of the preinitiation complex, and/or the positioning of nucleosomes at the promoter of the downstream gene[2–5]. The potential for this type of TI may increase in genomes with higher gene density; shorter distances between genes would require more precise termination of upstream genes. Thus controlling elongation and termination at upstream genes may be key in preventing TI at downstream genes.

Accumulating evidence suggests important regulatory roles for Pol II pausing[6–8] in shaping the transcriptome. An example of Pol II pausing seen in metazoans is the accumulation of transcriptionally engaged Pol II 30–50 bp downstream of the transcription start site (TSS)[7]. This promoter-proximal pausing is often seen at developmentally regulated genes, where it may facilitate their rapid activation, and is mediated by the DRB Sensitivity-Inducing Factor (DSIF) and the Negative Elongation Factor (NELF) complexes[9]. Mapping of engaged Pol II in *Arabidopsis thaliana* and maize, in contrast, did not reveal patterns of Pol II accumulation in regions immediately downstream of TSSs[10,11]. Thus plants, which lack NELF homologs, do not appear to make significant use of promoter-proximal pausing. In a phenomenon known as 3′ Pol II pausing, however, a significant increase in Pol II is observed near the transcript end site (TES) of many genes[11]. The molecular mechanisms that give rise to 3′ Pol II pausing in plants, as well as its biological significance, are unclear.

To better understand the role of 3′ pausing, we investigated a three-member family of putative negative transcription elongation factors from Arabidopsis, which we have named BORDER (BDR1, BDR2, and BDR3) proteins. BDR proteins are enriched in intergenic regions and promote the 3′ pausing of Pol II for a large fraction of genes. This activity is especially important at closely spaced genes on the same strand (i.e., in tandem). In the *bdr1,2,3* mutant, 3′ pausing is reduced at upstream genes and Pol II occupancy shifts into the promoter regions of the downstream genes. While expression of the upstream gene is unaffected in the *bdr1,2,3* mutant, the shift in Pol II from the upstream gene into the promoter region of the downstream gene is coincident with reduced expression of the downstream gene. In this way, BDR proteins prevent TI between closely spaced tandem genes.

## Results

**BDR proteins resemble transcriptional elongation factors**. BDR proteins form a three-member family in *Arabidopsis* (BDR1 = At5g25520, BDR2 = At5g11430, BDR3 = At2g25640). Each BDR protein contains an SPOC domain, which is found in the SPEN family of transcriptional repressors, and a transcription elongation factor IIS (TFIIS) central domain (Fig. 1a, Supplementary Fig. 1)[12,13]. TFIIS contains three domains (I, II/central, and III) and acts as a positive elongation factor. During elongation, RNA Pol II frequently backtracks, such that it is no longer positioned at the 3′ end of the growing transcript. To restart elongation, the central domain of TFIIS binds to RNA Pol II, while domain III stimulates cleavage of the nascent transcript, thus providing a

new 3′ end for RNA Pol II[12,14–16]. The fact that BDR proteins do not contain domain I or III suggests that the BDR proteins are unlikely to have TFIIS-like activity.

Proteins with similar domain organization are found outside plants, with fungal and animal proteins often including an additional N-terminal PHD domain[17–19] (Fig. 1a). These include the mammalian proteins SPOCD1, PHF3, and DIDO1[17,20,21]. The best characterized is the yeast protein BYpass of Ess1 (Bye1), which contains a PHD domain in addition to its SPOC and TFIIS central domains. Bye1 is thought to act as a negative elongation factor and binds to Pol II through its TFIIS central domain and to histone H3 trimethylated on lysine 4 (H3K4me3) through its PHD domain[17,22]. Bye1 is enriched in the 5′ regions of genes[17,23], and consistent with a role in repressing Pol II elongation, Pol II occupancy in the 5′ regions of genes is reduced in the bye1 mutant, whereas Pol II occupancy is increased in gene bodies[22].

To investigate the function of BDR proteins in Arabidopsis, we obtained T-DNA insertional mutants in *BDR1* (*bdr1-1*), *BDR2*, (*bdr2-1*), and *BDR3* (*bdr3-1*). Single mutants did not show clear phenotypes; however, the *bdr1,2,3* triple mutant showed a short-root phenotype (Fig. 1b, c). Given the similarity between the BDR proteins and negative elongation factors[17,22], we speculated that the mutant phenotypes might be caused by increased transcriptional elongation. If this is the case, inhibiting elongation might attenuate the phenotype of the *bdr1,2,3* triple mutant. To test this hypothesis, we grew seedlings in the presence of a chemical inhibitor of transcription elongation, 6-Azauracil (6AU)[24] and examined root growth. In contrast to wild type, which showed a reduction in root growth when grown on 6AU, root length was partially rescued in *bdr1,2,3* mutant seedlings (Fig. 1b and Supplementary Data 1). We also tested a second chemical inhibitor of transcription elongation, mycophenolic acid (MPA)[24]. Similar to the results with 6AU, MPA had a slight negative effect on root growth in wild type (Fig. 1c and Supplementary Data 1); however, *bdr1,2,3* root length more than doubled when grown on MPA. These results suggest that the short-root phenotype may be due to increased transcriptional elongation in the *bdr1,2,3* background.

**BDR proteins are enriched at gene borders**. We performed chromatin immunoprecipitation followed by next-generation sequencing (ChIP-seq) to determine the localization of BDR1, BDR2, and BDR3 using MYC-tagged constructs driven by their respective endogenous promoters in the *bdr1,2,3* background. All three constructs rescued the short-root phenotype of *bdr1,2,3* (Supplementary Fig. 2A). ChIP-seq showed that BDR1 and BDR2 are mainly enriched at gene borders, with peak summits located a short distance upstream of TSSs and/or downstream of TESs (Fig. 2a, b). Because intergenic distances in *Arabidopsis* are relatively short (e.g., Fig. 2a), it is often not possible to unambiguously assign an intergenic peak to one of the two neighboring genes, but BDR peaks are nevertheless found between both converging and diverging gene pairs (Fig. 2a). In contrast to BDR1 and BDR2, which show roughly similar binding in TSS and TES regions, BDR3 showed a strong preference for TES binding (Fig. 2b). Among the three BDR proteins, BDR1 showed the highest ChIP-seq enrichment and BDR3 showed the lowest (Fig. 2b, note different *y* axis scales).

We defined 21,334, 11,997, and 12,178 peaks for BRD1, BDR2, and BDR3, respectively. Consistent with their greater amino acid sequence similarity (Supplementary Fig. 1), we found the greatest overlap in peaks between BDR1 and BDR2. Approximately 82% of BDR2 peaks overlapped with BDR1 peaks, whereas only 22% of BDR3 peaks overlapped with BDR1 (Fig. 2c). For all three BDR proteins, occupancy is correlated with the expression of the

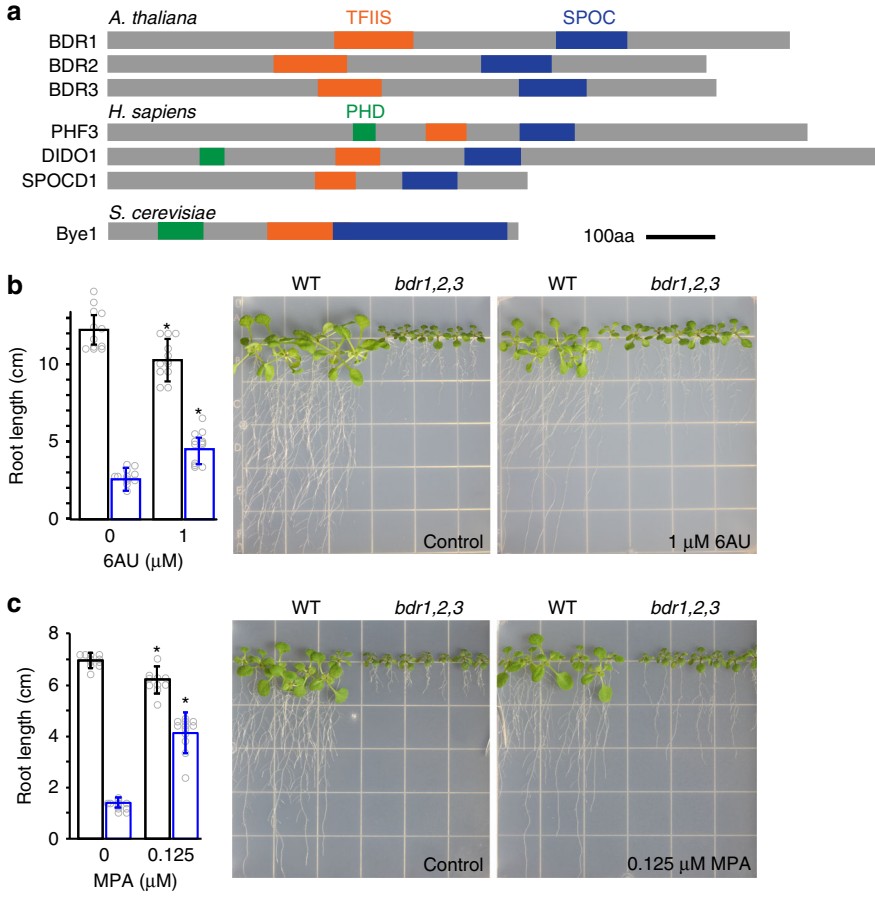

**Fig. 1** Reduced root growth in the *bdr1,2,3* mutant is partially restored by inhibitors of transcriptional elongation. **a** Schematic drawing of BDR proteins and related proteins from humans and yeast. **b, c** The short-root phenotype of *bdr1,2,3* is partially rescued by the transcription elongation inhibitors 6AU (**b**) and MPA (**c**). Wild type and *bdr1,2,3* are represented by black and blue bars, respectively ($n = 12$ biologically independent seedlings). Error bars indicate one standard deviation. Asterisks indicate significant differences ($p < 0.05$)

nearest gene (Fig. 2b). BDR1 and BDR2 peaks were enriched in the intergenic regions, such as promoters and regions immediately downstream of the TES, as well as 5′ untranslated regions (5′ UTRs; Fig. 2d). BDR3, in contrast, did not show enrichment in promoters or 5′ UTRs but was enriched in exons, 3′ UTRs, and regions immediately downstream of the TES. Because binding of BDR proteins is strongest near the TSS and/or TES, we examined occupancy in these regions in more detail. For each BDR protein, we identified sets of genes containing peaks within 300 bp of the TSS or TES and plotted the occupancy of the corresponding BDR protein (e.g., occupancy of BDR1 over TSS regions containing BDR1 peaks, occupancy of BDR2 over TSS regions containing BDR2 peaks, etc). In the TSS region, all three BDR proteins showed maximum occupancy slightly upstream of the TSS (Fig. 2e), with maxima of −87, −31, and −148 bp for BRD1, BDR2, and BDR3, respectively. BDR proteins showed less variation in binding position at the TES, with maxima of 106, 131, and 117 bp downstream of the TES for BRD1, BDR2, and BDR3, respectively. (Fig. 2f). BDR1 and BDR2 showed strong overlap in peaks at both the TSS and TES, whereas BDR3 showed stronger overlap with BDR1 and BDR2 at the TES than at the TSS (Fig. 2e, f).

BDR1 peaks are located in nucleosome-depleted, DNase-hypersensitive regions (Fig. 3a and Supplementary Fig. 3). Although less pronounced than for BDR1, regions with the highest occupancy for BDR2 and BDR3 also showed a preference for nucleosome-depleted regions (Supplementary Fig. 3).

Consistent with the correlation with gene expression (Fig. 2b), we found that the occupancy of all three BDR proteins also correlates with Pol II levels (Fig. 3a and Supplementary Fig. 3). Because BDR1, BDR2, and BDR3 show differences in binding in TSS and TES regions (Fig. 2b), we also examined the correlations between BDR proteins, Pol II, H3, and DNase-hypersensitive regions specifically at 250 bp regions immediately before the TSS, after the TSS, before the TES, and after the TES (Supplementary Fig. 4). These data show that some correlations are stronger in particular regions. For example, the correlation between BDR3, which shows relatively little binding near the TSS, with BDR1, BDR2, and Pol II is higher near the TES (Supplementary Fig. 4).

We also examined sequence conservation around BDR peaks. PhastCons[25] examines sequence conservation between *Arabidopsis* and the genomes of 20 other angiosperms. In order to focus on the conservation of intergenic regions, sequences corresponding to annotated genes were removed. Because BDR peaks are preferentially found in nucleosome-depleted regions, we included other nucleosome-free regions, as well as random intergenic sequences, as controls. We found that sequence conservation was significantly higher at BDR peaks compared to surrounding intergenic sequences (Fig. 3b), with higher conservation observed for BDR1 and BDR2 peaks than for BDR3 peaks. We also searched for overrepresented motifs in BDR1 and BDR2 peaks, focusing on the 101 bp surrounding the peak center. Two motifs were identified that occurred more frequently in BDR1 and BDR2 peaks than in other intergenic regions (Fig. 3c, d). A TCP-like

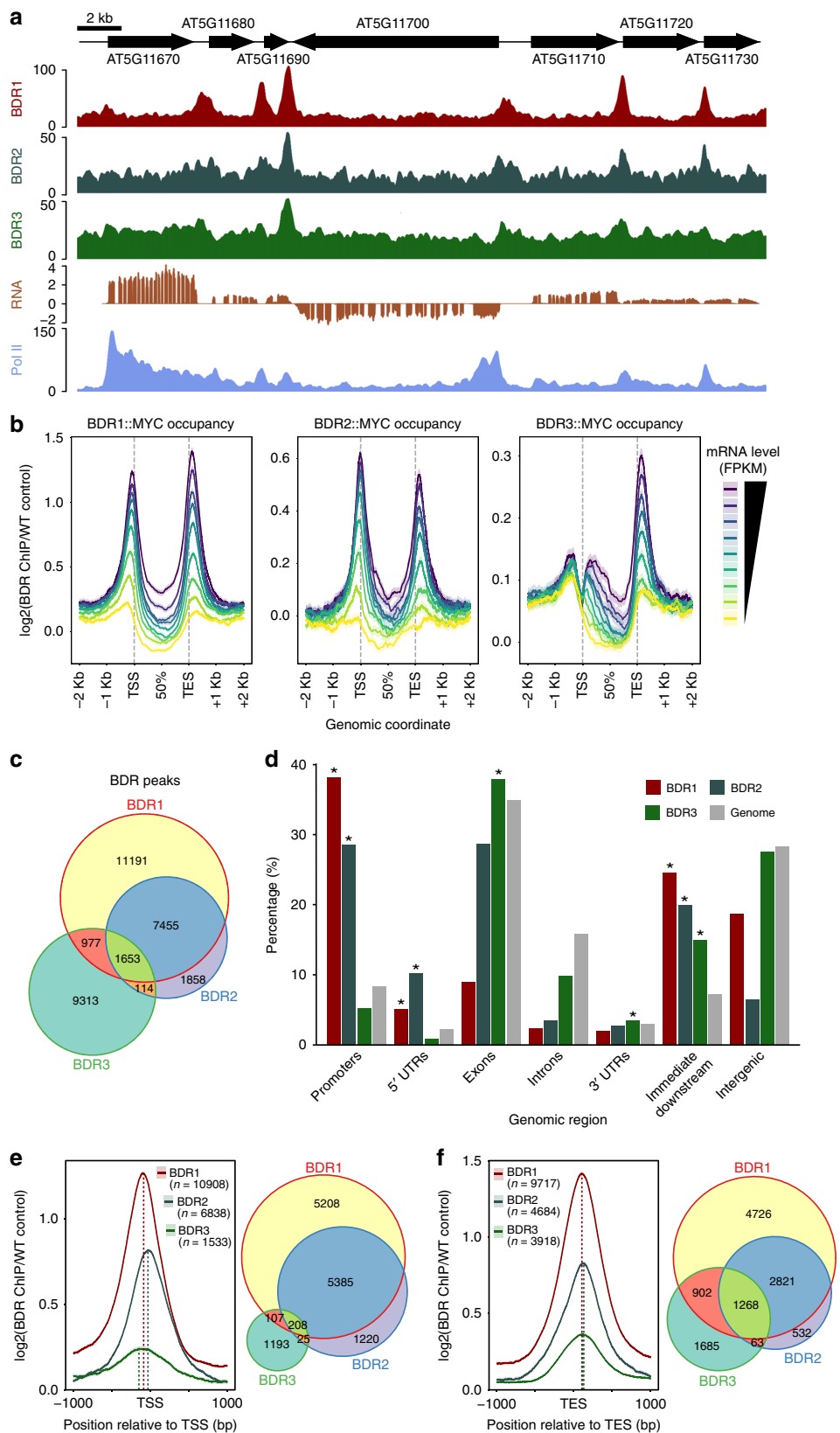

motif was found in 44.9% of BDR1 peaks and an E-box motif was found in 7.02%[26]. Both motifs were also enriched in BDR2 peaks (Fig. 3d). Interestingly, although BDR1 and BDR2 are enriched near both TSS and TES regions (Fig. 2b, e, f), these motifs only show enrichment near TSS sites (Fig. 3c). Because BDR proteins

lack characterized DNA-binding motifs, it is likely that recruitment to chromatin depends on interactions with other factors. The result that enriched sequence motifs are found near the TSS, but not near the TES suggests that BDR proteins may be recruited to chromatin through multiple interactions/mechanisms, e.g.,

**Fig. 2** Genome-wide localization of BDR proteins. **a** Browser track showing intergenic enrichment of BDR1, BDR2, BDR3, and Pol II. **b** Metagene profiles of BDR1::MYC, BDR2::MYC, and BDR3::MYC ChIP-seq signal in nine groups of genes defined by increasing mRNA expression levels in wild type. The average BDR ChIP-seq signal for each group (line) and the associated 95% confidence interval based on a Gaussian assumption (shade) are represented. Signal in gene bodies was averaged in bins of 1% of the gene size. FPKM fragments per kilobase per million aligned fragments. **c** Venn diagram showing the overlap between BDR1, BDR2, and BDR3 peaks. **d** Distribution of BDR ChIP-seq peaks in various classes of genomic features. Promoter regions and immediate downstream regions are defined as up to 300 bp upstream from the TSS or downstream of the TES, respectively. Intergenic regions are >300 bp from any gene. Asterisks indicate a significant enrichment compared to genome-wide distributions (p < 0.002). **e, f** Coverage of ChIP-seq signal for BDR1, BDR2, and BDR3 around the TSS (**e**) and TES (**f**). For each protein, genes were selected that contained peak summits <300 bp from their TSS (**e**) or TES (**f**). Venn diagrams illustrate the overlap between genes with BDR1, BDR2, and BDR3 peaks at their TSS (**e**) or TES (**f**)

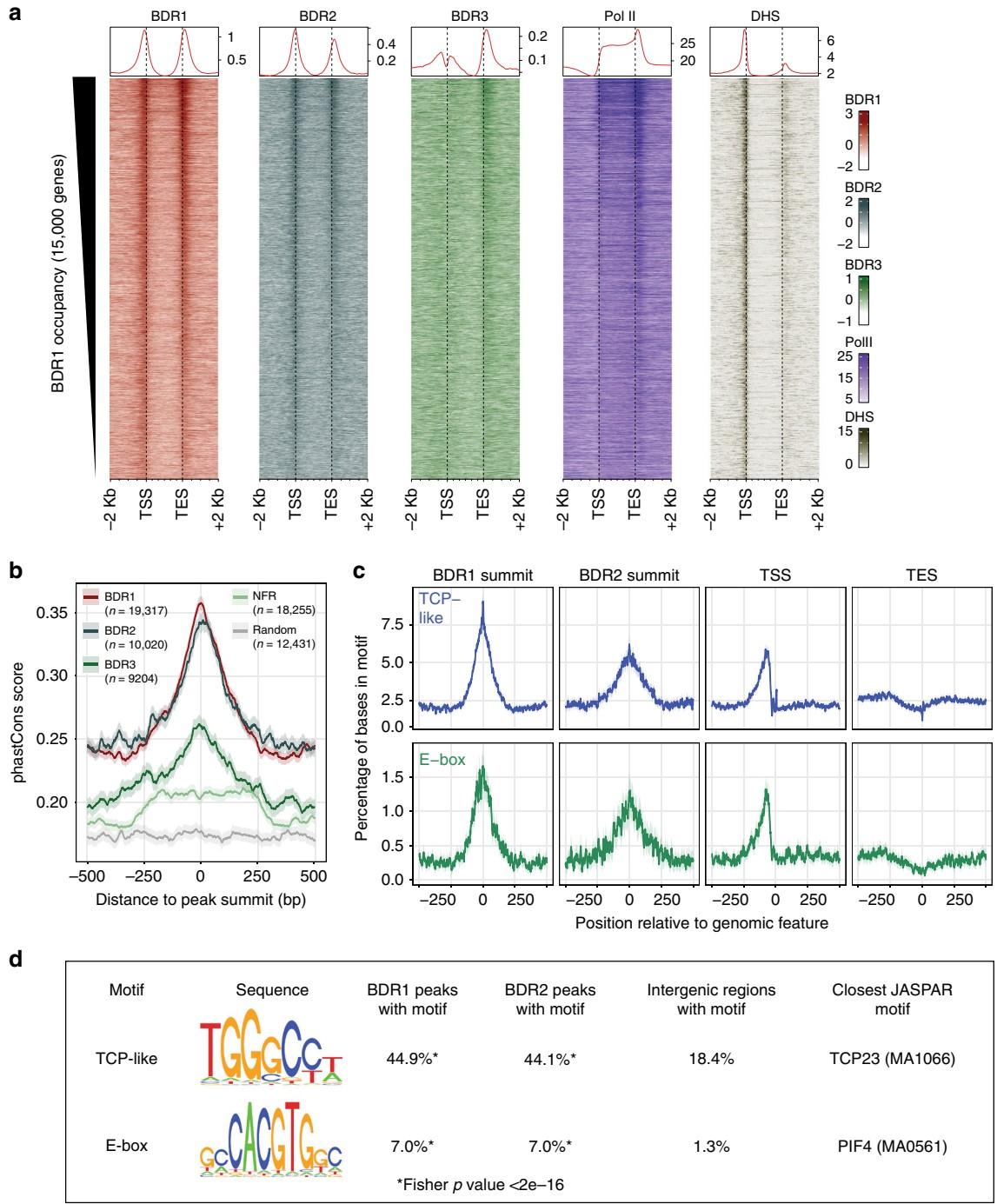

**Fig. 3** BDR peaks contain evolutionarily conserved TCP-like and E-box motifs. **a** Heatmap and metagene profiles (top) of ChIP-seq signals and DNAse-hypersensitive sites (DHS). Genes were sorted by total BDR1 signal around the TSS and TES; the top 15,000 genes are shown. **b** Sequence conservation across 20 angiosperms for intergenic regions around BDR1, BDR2, and BDR3 peak summits, nucleosome-free regions (NFR), or random regions. Average phastCons score (line) and 95% confidence intervals (shade) are shown. **c** Enrichment of TCP-like and E-box motifs in BDR1 and BDR2 summits, TSS, and TES regions. **d** Motifs identified in BDR1 and BDR2 summits

interacting with DNA-binding proteins that recognize TCP-like and/or E-box motifs near the TSS, and other proteins, such as components of the transcription termination machinery, near the TES. The model that BDR proteins may be recruited to TSS and TES regions through separate mechanisms is also supported by asymmetric binding profile of BDR3, which shows much stronger affinity for TES regions than for TSS sites (Fig. 2b).

**BDR proteins promote 3′ Pol II pausing.** The potential role of BDR proteins as negative transcription elongation factors and their enrichment near the 3′ ends of genes suggests that they may play a role in 3′ pausing. We determined Pol II occupancy in wild type and *bdr1,2,3* using antibodies recognizing Pol II, Serine 5 phosphorylated Pol II (S5P), and Pol II S2P. During transcription, Pol II undergoes a series of phosphorylation events, with Pol II S5P associated with initiation and Pol II S2P associated with elongation[27]. Consistent with this model, we observed that Pol II S2P signal increased through the body of the gene (Fig. 4a). S5P occupancy increased not only through the body of the gene but also showed a peak near the TSS and a depletion near the TES (Fig. 4a). Consistent with published ChIP-seq, GRO-seq, and pNET-seq studies[10,11,28], all three antibodies showed 3′ Pol II

accumulation just after the TES (Fig. 4a, red arrows), indicative of 3′ pausing.

We used ChIP-seq data from wild type and *bdr1,2,3* to calculate a 3′ pausing index for Pol II (ratio of read densities from the region immediately downstream of the TES to those of the gene body, Fig. 4b). We first examined the relationship between 3′ pausing and gene expression. In wild type, the level of 3′ pausing was correlated with gene expression, with the most highly expressed genes having the highest levels of 3′ pausing (Fig. 4c). In *bdr1,2,3*, 3′ pausing was significantly reduced for nearly all combinations of antibody and gene expression group (Fig. 4c). Thus BDR proteins do indeed promote 3′ pausing for a large fraction of genes.

**BDR-protected genes occur in a specific genomic context.** We used RNA-seq analysis to identify three sets of genes whose expression is promoted or repressed by BDR proteins (i.e., show decreased or increased expression in *bdr1,2,3* seedlings, respectively), as well as non-differentially expressed genes (Supplementary Fig. 5A). Interestingly, we found that BDR-promoted genes, which we will refer to as BDR-protected genes, preferentially occur in a specific genomic context (Fig. 5a and

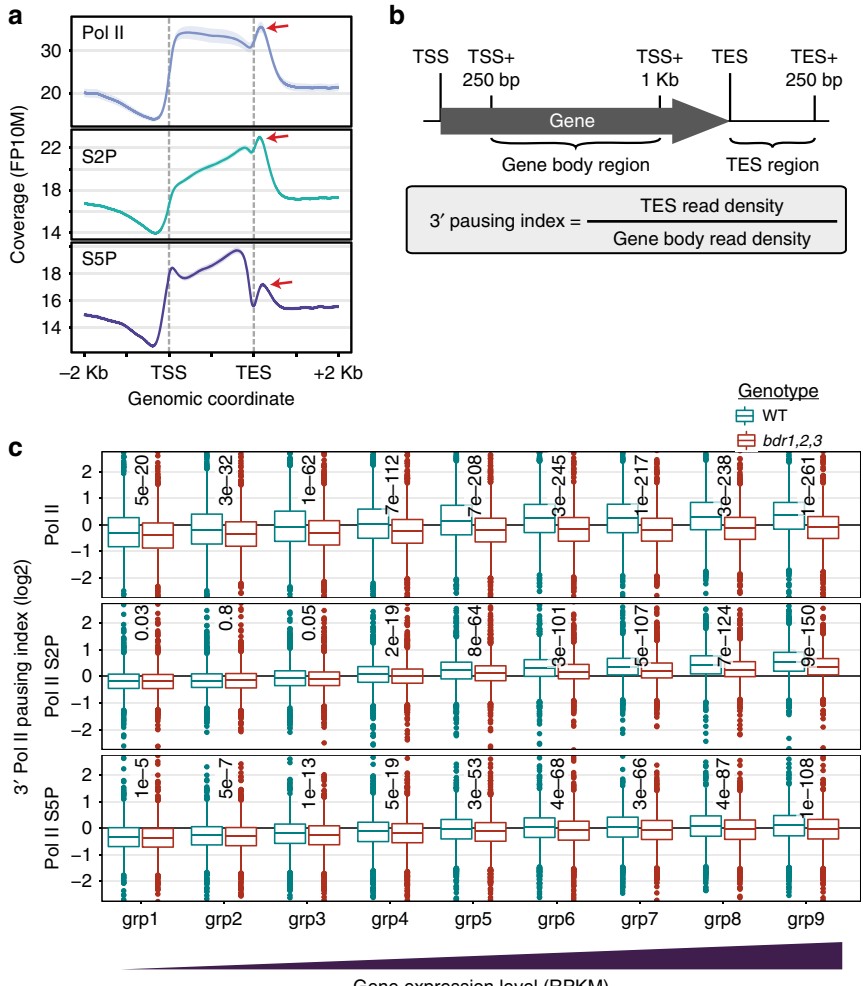

**Fig. 4** BDR proteins promote 3′ pausing and gene expression in a specific genomic context. **a** Pol II ChIP-seq coverage across expressed genes in *Arabidopsis* seedlings. 3′ pausing indicated by red arrows. **b** Calculation of a 3′ pausing index. **c** 3′ pausing indices for nine groups of genes defined by increasing mRNA expression levels in wild type. 3′ pausing is reduced in the *bdr1,2,3* mutant, particularly at highly expressed genes. The centerline of boxplots is the median. The bounds of the box are the first and third quartiles (Q1 and Q3). Whiskers represent data range but are bounded to 1.5-fold the interquartile range (Q3–Q1); points outside this range are represented individually

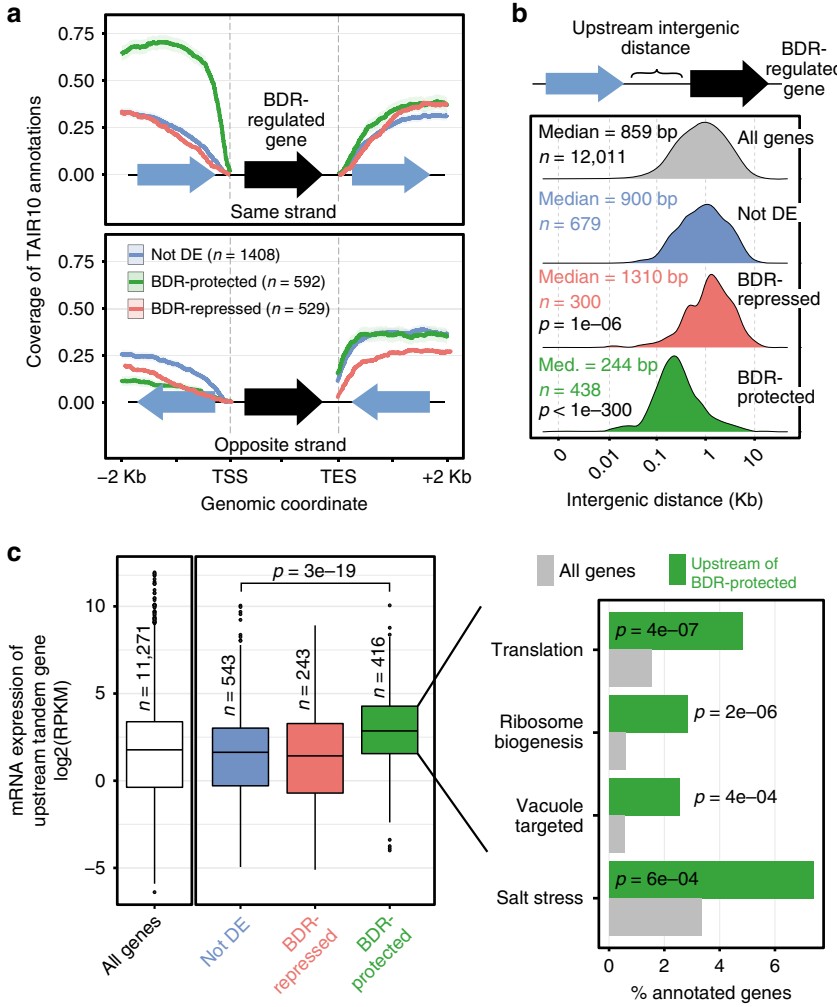

**Fig. 5** Genes positively regulated by BDR proteins tend to have highly expressed upstream neighbors on the same strand. **a** Fraction of BDR-regulated genes that have upstream or downstream neighbors at various orientations, within the indicated distances. **b** BDR-protected genes and their upstream tandem neighbors have short intergenic distances. Distribution of intergenic distances between tandem genes. Distribution differences relative to all genes were evaluated by Kolmogorov–Smirnov test with BH correction. **c** The upstream tandem neighbors of BDR-protected genes have relatively high expression levels. Wild-type gene expression levels for upstream tandem neighbors from the indicated groups of genes (left panel). Differences are evaluated by Mann–Whitney $U$ test. Right panel, gene ontology analysis (goseq R package) of the upstream tandem neighbors of BDR-protected genes. Categories with $p < 0.001$ are shown. The centerline of the boxplot is the median. The bounds of the box are the first and third quartiles (Q1 and Q3). Whiskers represent data range but are bounded to 1.5-fold the interquartile range (Q3–Q1); points outside this range are represented individually

Supplementary Fig. 5B). In all, 74% of BDR-protected genes are on the same strand as their immediate upstream neighbor, compared to 50% for all genes in the genome (Fig. 5a and Supplementary Fig. 5B). No significant enrichment for orientation was found in the downstream neighbor of BDR-protected genes, although BDR-repressed genes showed a slight preference for having a downstream tandem neighbor (Supplementary Fig. 5B). In addition to orientation, we also examined the intergenic distances between BDR-protected genes and their upstream neighbors. The TES of the upstream gene is much closer to the TSS of BDR-protected genes (244 bp) compared to the genome-wide median of 859 bp (Fig. 5b). Finally, we examined the expression levels of tandem upstream genes. The tandem upstream neighbors of BDR-protected genes were more highly expressed (~2.5-fold higher) than the tandem upstream neighbors of non-differentially expressed controls or BDR-repressed genes (Fig. 5c). These tandem upstream genes were enriched for functions related to protein translation, subcellular targeting, and salt stress

(Fig. 5c). In contrast to the BDR-protected genes themselves, the tandem upstream neighbors of BDR-protected genes were typically not differentially expressed in the *bdr1,2,3* triple mutant. Thus BDR-protected genes preferentially occur a short distance downstream of a highly expressed gene on the same strand and BDR proteins are required to maintain the expression of the downstream gene but not the upstream neighbor.

**BDR proteins promote 3′ Pol II pausing.** Given that BDR-protected genes are generally located a short distance downstream of a highly expressed neighbor on the same strand, we speculated that 3′ pausing at the upstream gene might be important in protecting the downstream gene from TI. TI is broadly defined as the direct negative impact of one gene's transcription on a second gene that is located in *cis*[1]. For example, it is possible for elongating Pol II from one gene to intrude into the promoter of a downstream gene on the same strand, disrupting its expression[2–5].

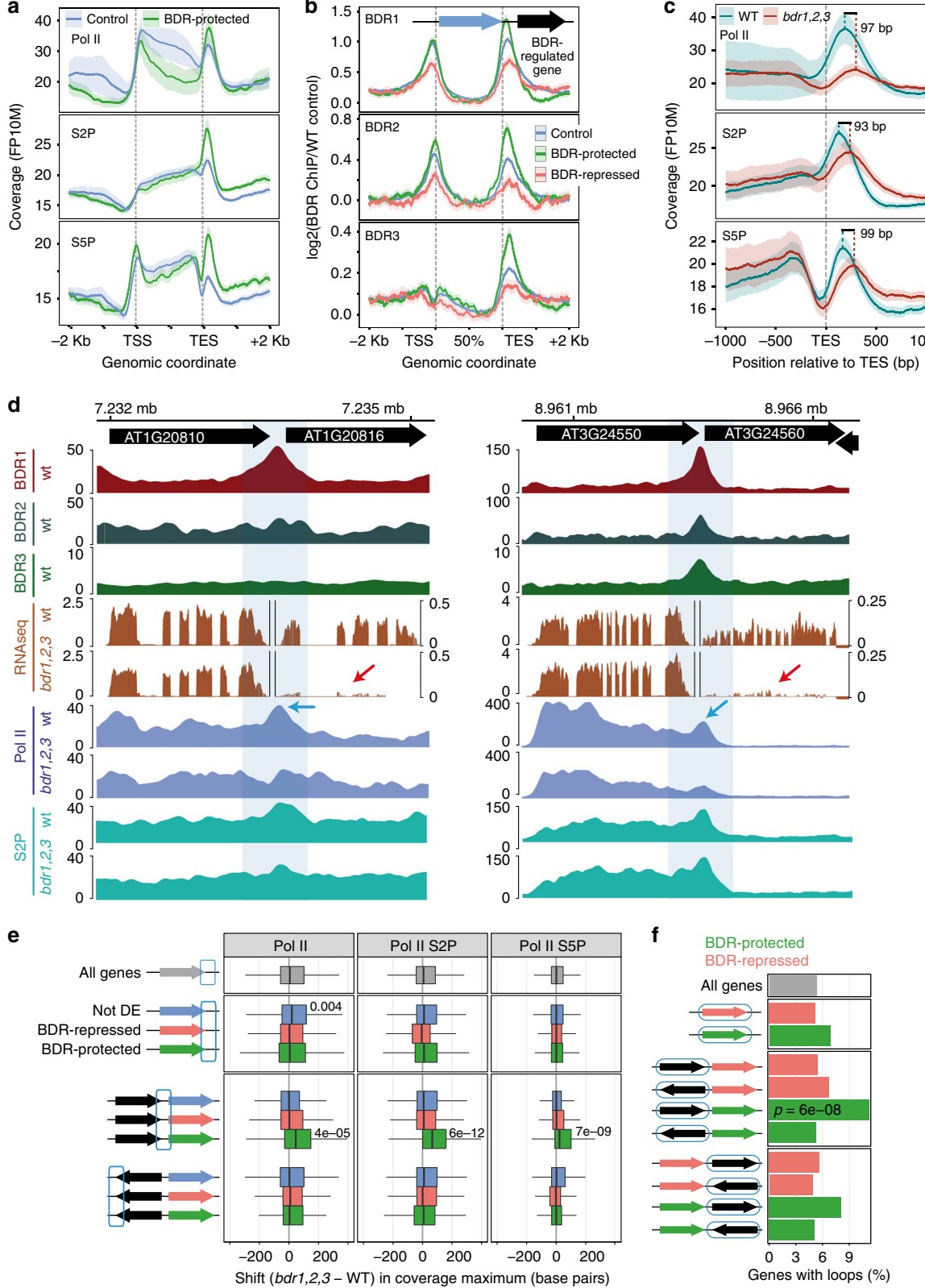

To explore the model that BDR proteins might be important in promoting 3′ pausing at the upstream neighbors of BDR-protected genes, we examined the Pol II occupancy at the upstream neighbors of BDR-protected genes in wild type. We found that the upstream neighbors of BDR-protected genes have elevated 3′ pausing compared to a set of 1500 control genes with

levels of expression similar to the upstream neighbors of BDR-protected genes (Fig. 6a and Supplementary Fig. 6). To determine whether the increased 3′ pausing is correlated with BDR proteins, we examined BDR protein occupancy at the upstream neighbors of BDR-protected genes. Compared to BDR-repressed genes or expression-matched control genes, BDR protein occupancy was

**Fig. 6** BDR proteins protect downstream genes from TI. **a** Levels of 3′ paused Pol II are elevated at tandem upstream neighbors of BDR-protected genes compared to expression-matched control genes. Metagene profiles of Pol II, Pol II-S5P, and Pol II-S2P ChIP-seq coverage across expressed genes in *Arabidopsis* seedlings. **b** BDR1 and BDR2 are enriched in the intergenic region between BDR-protected genes and their upstream tandem neighbors. Metagene profiles of ChIP-seq coverage at genes located upstream, on the same strand, as BDR-protected, BDR-repressed, or expression-matched control genes. **c** 3′ paused Pol II at upstream genes is reduced and shifted downstream in the absence of BDR proteins. Pol II near the TES of tandem upstream neighbors of BDR-protected genes is shifted ~96 bp downstream in the *bdr1,2,3* mutant. Average Pol II ChIP-seq profiles are presented. **d** Browser tracks of BDR-protected genes and their upstream tandem neighbors. Note the high BDR1 and BDR2 occupancy in the intergenic region, reduction in the expression of the downstream gene in the *bdr1,2,3* mutant (red arrows), and the reduction in 3′ paused Pol II at the upstream gene (blue arrows). **e** The downstream shift in the position of 3′ pausing in *bdr1,2,3* occurs preferentially at the upstream tandem neighbors of BDR-protected genes. The centerline of boxplots is the median. The bounds of the box are the first and third quartiles (Q1 and Q3). Whiskers represent data range but are bounded to 1.5-fold the interquartile range (Q3-Q1); points outside this range are omitted. **f** Upstream tandem neighbors of BDR-protected genes are enriched in gene loops[36]. Enrichment of gene loops in BDR-regulated genes and their neighbors. Statistics reflect the presence of loops in the circled gene in each context. *p* Values are shown for Fisher exact test with BH *p* value correction

higher at the upstream neighbors of BDR-protected genes (Fig. 6b), particularly near the TES. Because BDR occupancy is correlated with gene expression levels and Pol II occupancy (Figs. 2b and 3a and Supplementary Fig. 3), we investigated whether Pol II occupancy could account for this enrichment. Even after normalization of BDR ChIP-seq coverage by Pol II occupancy, the enrichment in BDR1 and BDR2 binding at tandem genes upstream of BDR-protected genes is still apparent (Supplementary Fig. 7).

We also examined Pol II occupancy in the *bdr1,2,3* mutant background. In the absence of the BDR proteins, the magnitude of 3′ pausing at the upstream neighbors of BDR-protected genes is reduced (Fig. 6c). We also found that peak 3′ pausing shifted ~96 bp toward the TSS of downstream BDR-protected genes (Fig. 6c). Because most BDR-protected genes are <250 bp away from their upstream tandem neighbor, 96 bp represents a significant fraction of the intergenic distance and has the potential to interfere with transcription initiation of the downstream gene or result in readthrough transcription. To investigate the latter possibility, we looked for evidence of chimeric readthrough transcripts, which have been reported in mutants for the RNA-binding protein *fpa*[29]. We saw no evidence, however, of chimeric transcripts between BDR-protected genes and their upstream tandem neighbors (Fig. 6d and Supplementary Fig. 8). This suggests that the reduced expression of downstream tandem genes in *bdr1,2,3* may be due to a failure of transcription factors to assemble at the promoter. Interestingly, while BDR proteins affect the magnitude of 3′ pausing for a large fraction of the genome (Fig. 4c), a significant shift in the position of 3′ pausing was consistently observed only for the upstream neighbors of BDR-protected genes (Fig. 6e).

**BDR proteins are correlated with gene loops**. Genome-wide application of chromatin conformation capture-based methods, such as HiC, revealed that large topologically associating domains (TADs) are a prominent feature of most eukaryotic genomes[30–33]. *Arabidopsis* was thought to be an exception to this trend, as large-scale TADs were not detected[34]. This initial observation, together with the lack of homologs of loop-forming insulator proteins such as CCCTC-binding factor (CTCF)[35], suggested that *Arabidopsis* may not make significant use of chromatin loops. Analyses performed at higher resolution, however, showed that short-range interactions are a major structural feature of the *Arabidopsis* genome. Approximately 1800 intragenic interactions have been identified between the 5′ and 3′ ends of genes[36]. To determine whether these "gene loops" might be associated with BDR protein function, we looked for an enrichment of loops in BDR-regulated genes. In BDR-protected and BDR-repressed genes, gene loops were not significantly overrepresented (Fig. 6f). Given

the evidence that BDR proteins aid in preventing TI in particular genomic contexts, we also determined the frequency of gene loops in the upstream and downstream neighbors of BDR-regulated genes. A significant enrichment in gene loops was only observed for the tandem upstream neighbors of BDR-protected genes (Fig. 6f). This correlation suggests that chromatin architecture may play a role in preventing TI.

**BDR proteins attenuate TI in response to photomorphogenesis**. If reduced gene expression in *bdr1,2,3* is the result of TI from upstream genes on the same strand, then increasing or decreasing the expression of upstream genes might exacerbate or relieve TI, respectively. To explore this possibility, we examined the changes in gene expression that occur at closely spaced tandem genes during photomorphogenesis, which results in the differential expression of a significant fraction of the genome. Wild-type and *bdr1,2,3* seedlings were grown for 4 days in the dark. On the fifth day, seedlings were either maintained in darkness or transferred to white light for the final 2 or 4 h prior to RNA isolation (Fig. 7a). Gene expression changes in response to light were largely similar between *bdr1,2,3* and wild type (Fig. 7b, c).

To look for evidence of increased TI when upstream genes are upregulated by light, we selected tandem genes with intergenic distances <600 bp, where expression of the upstream gene was similarly upregulated by light in both *bdr1,2,3* and wild type. We then determined the ratio of expression levels (*bdr1,2,3*/wt) for the downstream genes under dark and light conditions. Consistent with our model, we found that expression of the downstream gene was significantly reduced in *bdr1,2,3* upon upregulation of the upstream gene by light (Fig. 7d). We also found that TI could be relieved via the downregulation of upstream genes. Downstream genes that showed potential TI under dark conditions (i.e., reduced expression in the *bdr1,2,3* mutant) showed a significant increase in expression when the upstream gene was downregulated by light (Fig. 7e). Taken together, these experiments show that the TI in *bdr1,2,3* can be modulated by changing the expression of the upstream gene. Thus, in wild type, BDR proteins help to ensure the stable expression of downstream genes as their upstream tandem neighbors undergo light-regulated changes in gene expression.

We also observed that BDR proteins contribute to the rapid activation of light-regulated genes that have nearby upstream neighbors on the same strand, regardless of whether the upstream gene is light regulated. Among all genes that were light induced in wild type, we observed significantly reduced expression in *bdr1,2,3* when the light-induced gene had an upstream neighbor on the same strand and <600 bp away (Fig. 7f). For example, biochemical pathway analysis of the genes showing reduced

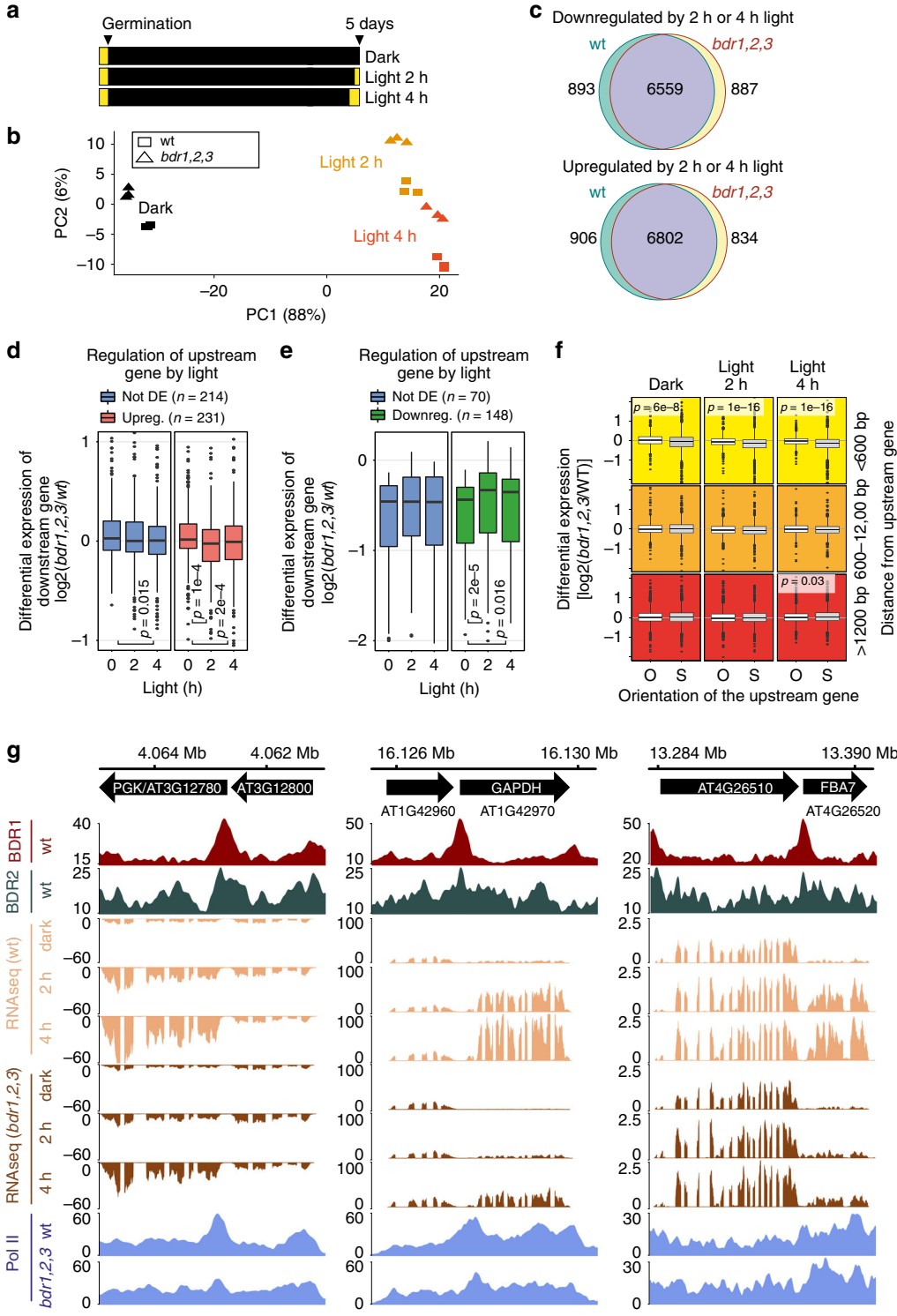

induction in *bdr1,2,3*, showed a significant enrichment for genes encoding Calvin–Benson–Bassham (CBB) cycle enzymes (Supplementary Fig. 9), which uses ATP and NADPH created by photosynthesis to convert carbon dioxide and water into organic compounds[37]. Three CBB cycle genes are located a short distance from an upstream gene on the same strand (Fig. 7g). Even though the upstream genes are not induced by light, the downstream CBB cycle genes show attenuated induction by light in the absence of BDR proteins (Fig. 7g).

## Discussion

TI between tandem genes was described in human alpha-globin genes >30 years ago[38] and similar examples have been reported in yeast[39,40], Drosophila[41], or following a T-DNA insertion in *Arabidopsis*[42]. At the genome-wide scale, however, our understanding of how often and to what degree TI might shape the transcriptome is still limited. Examples from yeast suggest that transcription-dependent changes in nucleosome occupancy and histone marks at the promoter of the downstream gene may

**Fig. 7** Modulation of TI during photomorphogenesis. **a** Schematic illustrating growth conditions used to generate 2 and 4-h light samples, as well as dark-grown controls. **b**, **c** Principal component analysis and Venn diagrams on RNA-seq data show the overall similar response of wild-type and *bdr1,2,3* mutant to light. **d** BDR proteins help prevent TI when nearby tandem upstream genes are upregulated by light. From all tandem gene pairs with an intergenic distance <600 bp, we selected those with upstream genes that were either upregulated or not differentially expressed at both 2 and 4 h light. Boxplots show decreased relative expression (log[*bdr1,2,3*/wt] values) of the downstream gene when the upstream gene is upregulated by light. Significant changes of log2(*bdr1,2,3*/wt) are evaluated by Wilcoxon signed-rank test. **e** Downregulation of the upstream tandem neighbor by light can attenuate TI in *bdr1,2,3*. From all tandem gene pairs with an intergenic distance <600 bp, we selected those with a downstream gene showing some evidence ($p < 0.05$) of transcriptional interference (i.e., downregulated in *bdr1,2,3* vs wt) under the darkness. Tandem pairs were then selected where the upstream gene showed either downregulation or no change in expression at either 2 or 4 h light. Boxplots show the upregulation of the downstream gene when the upstream gene is repressed by light (log[*bdr1,2,3*/wt] values). Significant changes of log2(*bdr1,2,3*/wt) are evaluated by Wilcoxon signed-rank test. **f** Light-upregulated genes show decreased expression in *bdr1,2,3* when located a short distance from an upstream neighbor on the same strand. Genes activated by light in wild type were binned by distance to their upstream neighbor. In each case, the expression ratio (*bdr1,2,3*/wt) was compared for genes having upstream neighbors on the same vs opposite strands using Mann–Whitney $U$ test with BH adjustment. The centerline boxplots is the median. The bounds of the box are the first and third quartiles (Q1 and Q3). Whiskers represent data range but are bounded to 1.5-fold the interquartile range (Q3–Q1); points outside this range are represented individually. **g** Browser tracks showing genes encoding three CBB cycle enzymes. *PGK*, *GAPDH*, and *FBA* are rapidly induced by light and are located a short distance from an upstream neighbor on the same strand. In *bdr1,2,3*, the upregulation by light is reduced

contribute to TI[2,43]. This does not appear to be the case for BDR-protected genes, however, as our MNase and histone ChIP-seq data indicate that the intergenic regions upstream from BDR-protected genes are nucleosome depleted. Rather, our data suggest that a reduction in 3′ Pol II pausing and a shift in Pol II occupancy at the upstream gene are likely responsible for perturbing the function of the downstream promoter in *bdr1,2,3* mutant. The precise mechanism by which BDR proteins promote 3′ Pol II pausing is unclear; however, it is tempting to speculate that an increased Pol II elongation rate in *bdr1,2,3* mutant might be responsible for a shift in Pol II termination site, as a downstream shift in termination has been observed using a "fast" elongating Pol II in human cells[44].

Although plants lack homologs of canonical animal insulator proteins, such as CTCF, the role of BDR proteins in ensuring that transcription of an upstream gene does not interfere with the expression of a closely spaced downstream neighbor can be thought of as a type of insulating activity. This suggests interesting parallels in the relationships between gene expression and chromatin organization in animals and Arabidopsis. In animals, it is common for enhancer elements to be located many kilobases away from the target gene[45]. This creates a twofold problem of how to ensure that an enhancer element associates with/promotes the expression of the correct gene, while making sure that it does not affect the expression of other nearby genes. In animals, CTCF and cohesin help to solve both problems through the formation of loops/TADs, where sequences inside the loop are more likely to interact with each other than with sequences outside the loop. In this way, enhancer elements preferentially associate with genes inside the same loop and are "insulated" from genes outside the loop[35]. Arabidopsis regulatory sequences, in contrast, are most often located near the promoter; examples of enhancer elements acting at a significant distance are rare[46]. Thus there may be less need for CTCF-type insulators and large-scale TADs. The relatively compact genome of Arabidopsis, however, creates other problems, such as TI between closely spaced genes. Interestingly, part of the solution in plants may also involve chromatin loops. The enrichment of gene loops and BDR proteins in upstream tandem genes suggests that they may play a role in promoting 3′ pausing and/or Pol II recycling[47] thereby preventing TI with downstream neighbors.

Taken together, these results indicate that BDR proteins inhibit TI by promoting 3′ pausing at upstream genes, thereby protecting the promoter region of the downstream gene from invasion by upstream, terminating Pol II. It is interesting to note that, although 3′ pausing is reduced at upstream genes in *bdr1,2,3*, the expression of the upstream genes themselves is usually not affected. This suggests that a gene's 3′ pausing may be more important for protecting the expression of its neighbors than for its own expression. This type of an activity would be predicted to be particularly important in an organism, such as Arabidopsis, with relatively short intergenic regions.

## Methods

**Plant material and growth conditions**. *bdr1-1*, SALK_142108C; *bdr2-1*, WISCD-SLOX352H03; and *bdr3-1*, SALK_059905C were obtained from the *Arabidopsis* Biological Resource Center (Columbus, OH) and confirmed by Sanger sequencing. Plants were grown at 22 °C in long days (16-h light/8-h dark) under cool-white fluorescent light with a light intensity of approximately 125 μmol m$^{-2}$ s$^{-1}$.

**Constructs**. For epitope tagged constructs, the *BDR1*, *BDR2*, and *BDR3* genomic DNAs without stop codons were transferred from pENTR to the destination vector pGWB16[48], which contains *4xMYC*. Resulting constructs were used as templates to amplify *BDR1::4×MYC*, *BDR2::4×MYC*, and *BDR3::4×MYC* using primers that incorporate Sbf I and Spe I sites (for *BDR1*: At5g25520-P1-sbf I-F cacctgcaggtc tctctttcccaaaaatttcaaaac+2701-pGWB16-myc-spe I-R actagtgatcggggaaattcgagctct aagcgctaccg; for *BDR2*: At5g11430-P1-Sbf I-F cacctgcaggatggccattgtttatttctaagg +2701-pGWB16-myc-spe I-R; for *BDR3*: At2g26540-P1-Sbf I-F cacctgcaggacttttg atatatccaaagggaattcg+2701-pGWB16-myc-spe I-R). The resulting fragments were first cloned into pENTR/D-TOPO and subcloned between Sbf I and Spe I sites in pMDC30[49].

**RNA expression analysis**. For RNA-seq, total RNA was isolated from *Arabidopsis* seedlings using the Trizol reagent (Sigma) or Spectrum Plant Total RNA Kit (Sigma) or Plant/Fungi Total RNA Purification Kit (Norgen) following the manufacturer's instructions. The integrity of RNA samples was analyzed with Agilent Technologies 2200 Tape Station (Agilent Technologies). Input was quantified by using the Qubit RNA BR Assay Kit. RNA-seq Libraries were prepared from total RNA using the TruSeq Stranded mRNA LT Sample Prep Kit (Illumina) and were sequenced on an Illumina NextSeq 500 at the Center of Genomics and Bioinformatics, Indiana University or Illumina Hiseq 2000 at the Genome Sequencing Facility in the Greehey Children's Cancer Research Institute of University Texas Health Science Center, San Antonio. All high-throughput sequencing data and corresponding experimental details are available in GEO SuperSeries GSE112443.

**Chromatin immunoprecipitation followed by next-generation sequencing**. Nuclei were isolated from cross-linked samples as described previously[50] and resuspended in nuclei lysis buffer (50 mM Tris-HCl pH8, 10 mM EDTA, 1% sodium dodecyl sulfate (SDS), 1 mM phenylmethanesulfonylfluoride (PMSF), 1% Plant Protease Inhibitors from Sigma). After fragmentation using a Covaris S200, the chromatin samples were diluted with ChIP dilution buffer (final concentration: 1% Triton X-100, 2 mM EDTA, 20 mM Tris-HCl pH8.0, 150 mM NaCl, 1 mM PMSF, 0.1% SDS, 1% Plant Protease Inhibitors, Sigma). The diluted chromatin samples were subjected to immunoprecipitation with antibodies (anti-MYC tag, clone 4A6, Millipore 05–724 (30 μg); Anti-RNA polymerase II CTD repeat YSPTSPS antibody [8WG16] Abacm ab817 (20 μg); Anti-RNA polymerase II CTD repeat YSPTSPS (phospho S2) antibody, Abcam ab5095 (30 μg); Anti-RNA polymerase II CTD repeat YSPTSPS (phospho S5) antibody Abcam ab5131 (30 μg); and control IgG Abcam ab18413 (20 μg)).

Native histone ChIP was performed as described previously[51] using anti-Histone H3 Abcam ab1791 (10 μg).

The ChIP libraries were prepared using the NEBNext® Ultra™ DNA Library Prep Kit (New England Biolabs) and then sequenced on a NextSeq 500 (Illumina)

at the Center of Genomics and Bioinformatics, Indiana University. All high-throughput sequencing data and corresponding experimental details are available in GEO SuperSeries GSE112443.

**RNA-seq computational analysis.** Two independent RNA-seq studies each with biological triplicates were performed in wild type and *bdr1,2,3* triple mutant seedlings that were grown under standard conditions (GSE112440 and GSE112441). The second study (GSE112441) also included *bdr1*, *bdr2*, and *bdr3* single mutant seedlings. Except when otherwise stated, study GSE112441 was used to compute the figures presented in this manuscript. We systematically verified that consistent results were obtained with both the RNAseq studies.

For GSE112440 (49 bp single-end reads sequenced on Illumina HiSeq 2000 instrument) and GSE112441 (2 × 43 bp paired-end reads sequenced on Illumina NextSeq 500 instrument), read alignments (For GSE112440: topHat 2[52]; for GSE112441: STAR[53]), filtering to keep uniquely aligned reads (samtools and grep commands[54]), and gene-level read counting (featureCounts[55] and differential expression analysis with DESeq2[56]) were performed as described in the corresponding GEO records. Genes with Benjamini–Hochberg (BH)-adjusted $p$ values <0.05 were considered differentially expressed. Using GSE112441 RNA-seq data, we defined a list of 1408 control, non-differentially expressed genes ("Not DE") by selecting genes with high $p$ values ($p > 0.45$) and low absolute log2(fold-change) (<0.25) for all comparisons (single *bdr1*, *bdr2*, and *bdr3* mutants and the *bdr1,2,3* triple mutant vs wild type) and removing genes with extreme read counts (DESeq2 basemean >3 and <1e5). Enrichments for Gene Ontology (GO) biological processes among the gene sets (upregulated or downregulated) were evaluated using the goseq package[57].

**ChIP-seq computational analysis.** All ChIP-seq samples were sequenced in paired-end mode on an Illumina NextSeq500 instrument (read length of 40, 43, or 155 bp, as specified in the corresponding GEO entries). Sequencing adapters were removed using Trimmomatic 0.33 in paired-end mode[58] and reads were aligned to the *Arabidopsis* genome using Bowtie2[59] using the –dovetail parameter and a maximum insert size of 1 Kb. Duplicate fragments were removed with Picard 2.2.4 MarkDuplicates (http://broadinstitute.github.io/picard/). Samtools v 1.3 was used to keep only reads mapped in proper pairs with mapping quality (MapQ) >2. For MNase-seq and ChIP-seq on histone modifications, we found that fragments <70 bp were enriched for background signal and fragments >250 bp for signal corresponding to dinucleosomes. Thus we only kept the reads corresponding to fragment sizes between 70 and 250 bp. Aligned reads were imported in R (v.3.3.2) to obtain coverages using Bioconductor v3.4[60]. Coverages were normalized as fragments per 10 million fragments (FP10M) and exported to bigWig files using the rtracklayer package[61]. ChIP-seq peaks were detected using MACS2 2.1.0[62] in paired-end mode. Peaks located in blacklisted regions were removed. Annotation of peaks relative to genomic features were obtained using the ChIPpeakanno package[63].

**Average profiles and metagene plots.** Coverages (e.g., FP10M for ChIP-seq data, phatsCons score, or annotation coverages) or normalized coverages (e.g., log2(BDR ChIP/WT control ChIP)) were directly used, without binning or smoothing, to produce average profiles centered on genomic features of interest (e.g., peak centers, TSS, or TES). After selecting a gene set of interest, the most extreme 0.01% coverage values were replaced by the upper 99.99th percentile value before calculating the average value ("metagene", solid line) and the associated 95% confidence interval (CI) (shade). For the latter, we used a normal approximation, which on several examples gave results nearly identical to bootstrap estimates of the CIs as implemented in the ChIPseeker package[64]. For metagene plots centered on gene bodies, we first averaged the signal in 100 bins covering the gene body (i.e., bin size of 1% of gene length).

**Multigene heatmaps.** Multigene heatmaps were produced with the EnrichedHeatmap package[65] from coverages (in FP10M) or normalized coverages that were averaged in 20 bp bins before/after genomic features of interest (TSS, TES, or peak center) or in bins covering every 1% of gene length along gene bodies.

**Definition of a blacklist for the *Arabidopsis* genome.** We used 20 control samples (input DNA or IgG ChIP) obtained in our laboratory from different ChIP-seq experiments (both sonicated and MNase-fragmented chromatin) to define a blacklist of genomic regions with systematically high signal in control samples. We used the Bioconductor package GreyListChIP[66] to generate a list of regions with high signal (95th percentile of negative binomial distribution estimated from 100 random samples of size 30,000) in >50% of the control samples and refined these regions manually on a genome browser using independent ChIP and control samples. A bed file of blacklisted regions is provided as a supplementary file (BlackList_TAIR10.bed).

**Bioinformatic methods for each figure.** Figure 2a. Coverages from BDR1::MYC (GSE113059), BDR2::MYC (GSE113059), BDR3::MYC (GSE131772), and Pol II (GSE113078) ChIP-seq fragments (units: FP10M) and average coverage from

RNA-seq (GSE112441) fragments obtained from 3 wild-type samples (units: RPM, sign indicating on which strand the reads align) were plotted for ~32 Kbp region of chromosome 5 using the Gviz package[67].

Figure 2b. Metagene profiles of BDR1::MYC, BDR2::MYC, and BDR3::MYC normalized ChIP-seq signal were obtained for 9 groups of genes defined by increasing mRNA expression levels in wild type ($n$ = 2232–3005 genes per group, Supplementary Data 1, Table S1). The average BDR-normalized ChIP-seq signal for each group (line) and the associated 95% CI based on a Gaussian assumption (shade) are represented. Signal in gene bodies was averaged in bins of 1% of the gene size. For each group, we obtained metagene profiles using the ChIP-seq data for BDR1::MYC (left), BDR2::MYC (center), or BDR3::MYC (right) as described above.

Figure 2c. ChIP-seq peaks for BDR1::MYC, BDR2::MYC, and BDR3::MYC were identified using MACS 2.1.0[62] in paired-end mode using the corresponding control ChIP performed on the same day in wild-type plants using the same anti-MYC antibody. After removing peaks located in blacklisted regions, we obtained 21,334 peaks for BDR1, 11,997 peaks for BDR2, and 12,178 peaks for BDR3 ($p < 0.01$, peak coordinates available in GSE113059 and GSE131772). Overlaps between the peaks were evaluated with the ChIPpeakAnno package[63] and plotted as a Venn diagram using the Vennerable R package[68].

Figure 2d. The summits of BDR1, BDR2, and BDR3 peaks were annotated relative to genomic features with the ChIPpeakanno package[63] using the following order of precedence (e.g., once a peak is annotated as promoter, it cannot be attributed to another category): Promoters (<300 bp upstream of TSS), immediate downstream (<300 bp downstream of TES), 5′-UTR, 3′-UTR, exons, introns, intergenic regions (>300 bp from any gene annotation). The percentage of peaks annotated in each category is shown. For comparison, the distribution of each genomic feature in the whole genome (using the same annotation precedence) is also illustrated (gray bars). Asterisk (*) indicates a significant enrichment of peaks within the corresponding annotation. Statistical significance of the enrichment for each genomic feature was assessed using 10,000 random samples of genomic positions.

Figure 2e. For each BDR protein, we identified all the protein-coding genes having a BDR peak within 300 bp of their TSS and plotted the average normalized coverage of the corresponding BDR::MYC ChIP. Maximum normalized coverage were found at −87, −31, and −148 bp upstream of the TSS for BDR1, BDR2, and BDR3, respectively. The Venn diagram illustrates the overlap between the gene lists obtained for the different BDR proteins.

Figure 2f. As for Fig. 2e, we identified for each BDR protein the protein-coding genes with a BDR peak within 300 bp of their TES and plotted the corresponding average profiles around the TES and the Venn diagram illustrating the intersections of gene lists. Maximum normalized coverages were found at +106, +131, and +117 bp downstream of the TES for BDR1, BDR2, and BDR3 respectively.

Figure 3a. Coverages (normalized by WT ChIP for BDR proteins) were averaged in 20 bp bins before the TSS and after the TES, and in bins of 1% of gene length along the gene bodies. For each gene, we summed BDR1 signal at the TSS ±10 bins and at the TES ±10 bins to obtain gene-specific BDR1 occupancy. The top 15,000 genes in terms of BDR1 occupancy were sorted in decreasing order to draw heatmaps using the EnrichedHeatmap package[65]. DNase-hypersensitivity signal was obtained from GSE34318[69].

Figure 3b. Sequence conservation (phastCons) scores from the alignment of 20 angiosperm plant genomes were obtained from ref. [25] and values overlapping with TAIR10 gene annotations were masked in order to focus on intergenic sequence conservation only. We selected:

1. 19,317 (BDR1), 10,020 (BDR2), or 9,204 (BDR3) ChIP-seq peaks that overlapped to some extent with intergenic regions;
2. 12,431 random regions overlapping with intergenic regions but not overlapping with BDR1 peaks or with blacklisted regions; and
3. 18,255 nucleosome-free regions defined as regions with >500 bp of consecutive MNase-seq and H3 ChIP-seq coverage below the first quartile of the respective datasets (GSE113076) and not overlapping with BDR1 peaks or blacklisted regions.

For each of these groups, we plotted the average phastCons scores around the respective BDR peak summits (±500 bp) or the respective region centers. Shades around the lines show the associated 95% CI using a Gaussian approximation.

Figure 3c. For the two motifs that were found to be enriched under BDR1 and BDR2 peaks (see Fig. 3d), the TCP-like motif and the E-box motif, we plotted their distribution around BDR1 or BDR2 peak summits and around the TSS or TES of protein-coding genes. At each base, the plot represents the percentage of bases that overlap with a TCP-like (blue) or an E-box motif (green), along with the 95% CI based on a Gaussian assumption (shade).

Figure 3d. We performed a de novo search for motifs enriched under BDR1 and BDR2 peaks using the peak-motif program from Regulatory Sequence Analysis Tools: RSAT[70]. We found highly similar motifs using BDR1 or BDR2 peaks so we only present the results for BDR1. We resized BDR1 peaks to ±50 bp around their summit and kept the peaks contained within intergenic regions (12,573 BDR1 peaks). We used control 101 bp intergenic regions with low nucleosome-associated signals (MNase and H3 ChIP), as defined for Fig. 3b, as background. Applying RSAT matrix-clustering, we identified two clusters of motifs corresponding to a TCP-like motif and an E-box motif. The core motifs from these two clusters were further trimmed to remove edges with low information content. We used the

resulting position-weight matrices to scan all intergenic regions and evaluate the enrichment of the motifs under intergenic BDR1 peaks using a Fisher exact test.

Figure 4a. For all the expressed genes, we plotted the metagene profiles obtained in wild-type samples for ChIP-seq performed with antibodies targeting the unphosphorylated C-terminal domain (CTD) of the large Rpb1 subunit of RNA polymerase II (PolII, abcam 817) or the CTD phosphorylated on the Ser2 (S2P, abcam ab5095) or Ser5 (S5P, abcam ab5131) residues of the YSPTSPS repeats. ChIP-seq data are from GSE113078.

Figure 4b. For each expressed gene >1 Kb, we calculated the Pol II 3′ pausing index as the ratio of read density just after the TES to read density in the gene body, as illustrated. Read density is the ratio of the number of reads aligned on the region to the region length in bp. The 3′ pausing indexes were calculated for wild type and *bdr1,2,3* using Pol II, S2P or S5P ChIP-seq data from GSE113078.

Figure 4c. We split genes into 9 groups according to their mRNA expression levels as in Fig. 2b and plotted as boxplots the distribution of log2 (3′ pausing indexes) for wild-type and *bdr1,2,3* plants using Pol II (upper panel), S2P (middle panel), or S5P (lower panel) ChIP-seq data from GSE113078. Significance of the difference in mean log2 (3′ pausing index) values between wild type and *bdr1,2,3* was evaluated using paired Student's *t* test with BH *p* value adjustment. Adjusted *p* values for all comparisons are shown.

Figure 5a. We calculated the strand-specific coverage of TAIR10 gene annotations and plotted the average coverage (line) and associated 95% CI (shade, normal assumption) around control, "Not DE" genes, and genes upregulated (BDR-repressed) or downregulated (BDR-protected) in the *bdr1,2,3* triple mutant compared to wild-type plants. Genes significantly regulated in *bdr1,2,3* mutant compared to wild type are illustrated in Supplementary Fig. 4A.

Figure 5b. Using the same gene groups as in Fig. 5a, we plotted the distribution of intergenic distances between the BDR-regulated genes and their upstream gene on the same strand. Significance of distribution differences was evaluated by Kolmogorov–Smirnov test with a BH correction. Only adjusted *p* values <0.01 are shown.

Figure 5c. From the gene groups defined for Fig. 5a, we selected the genes with an upstream gene on the same strand and a non-null read count in our RNA-seq study GSE112441. Then we plotted as boxplot the distribution of expression values (average log2(RPKM) from triplicate wild-type samples, GSE112441) for the upstream tandem genes and evaluated the significance of the differences between distributions using Mann–Whitney U test. Enrichment for GO biological processes among the upstream tandem neighbors of BDR-protected genes was evaluated with the goseq R package[57]. Genes located upstream on the same strand of all expressed genes were used as the gene universe. Only categories with a *p* value <0.001 are shown. Percentages of genes annotated with each GO category in the universe (gray) and in BDR-protected genes (green) are shown.

Figure 6a. We defined a set of 1500 control genes, located upstream, on the same strand of non-differentially expressed genes and matching the expression level of upstream tandem neighbors of BDR-protected genes (see Supplementary Fig. 5). For these 1500 control genes (blue line), and for genes located upstream, on the same strand of genes downregulated in the *bdr1,2,3* triple mutant (BDR-protected, *n* = 417 genes, green line), we plotted their metagene profiles obtained in wild-type samples using Pol II, S2P, and S5P ChIP-seq data from GSE113078.

Figure 6b. For control genes (*n* = 1500, blue line) and for genes upregulated (*n* = 529, BDR-repressed, red line) or downregulated (*n* = 592, BDR-protected, green line) in the *bdr1,2,3* triple mutant compared to wild-type plants, we selected the upstream genes that were located on the same strand and plotted the average normalized coverage (solid line) and associated 95% CI (shade) for BDR1::MYC, BDR2::MYC, and BDR3::MYC ChIP-seq fragments on their gene bodies and up to 2 Kbp on each side of gene borders.

Figure 6c. For genes downregulated in *bdr1,2,3* (BDR-protected genes, same as Fig. 6b), we identified the genes with an upstream gene on the same strand and plotted the average coverage (FP10M) of Pol II ChIP-seq data (antibodies as defined in Fig. 4a) obtained in wild type and the *bdr1,2,3* triple mutant at the TES (±1 Kbp) of these upstream genes. The shade represents the corresponding 95% CI. The distance between the maximum values of the average coverage for wild type and *bdr1,2,3* mutant is indicated.

Figure 6d. We used the Gviz Bioconductor package[67] to plot the coverage tracks for two genomic regions corresponding to genes downregulated in the *bdr1,2,3* triple mutant (BDR-protected) and their upstream tandem neighbor. BDR1::MYC and BDR2::MYC ChIP-seq data (FP10M) are from GSE113059, BDR3::MYC (FP10M) is from GSE131772, Pol II ChIP-seq data (FP10M) in wild-type and the *bdr1,2,3* triple mutant are from GSE113078, and RNA-seq data (RPM) are averages from triplicates of wild type or *bdr1,2,3* triple mutant samples from GSE112441. Note that for RNA-seq data we used two different scales for the regulated genes and its upstream neighbor due to the high expression of the latter. An additional nine regions are shown in Supplementary Fig. S7.

Figure 6e. For each antibody (columns), we plotted the distribution (boxplots) of shifts observed at the TES of different groups of genes (rows), either differentially expressed (DESeq2, false discovery rate (FDR) <5%) or not ("Not DE" controls) in *bdr1,2,3* compared to wild type or located directly upstream of these genes. For simplicity, "outliers" (points outside boxplot whiskers) are not represented. We used Wilcoxon signed-rank test with a BH-adjustment of *p* values to evaluate whether the shift was greater compared to the average shift observed for all expressed genes (gray boxplots). BH-adjusted *p* values <0.05 are represented.

Figure 6f. Using the same data as for Fig. 6e, we assessed by Fisher exact test the enrichment of genes forming gene loops in genes regulated in the *bdr1,2,3* mutant (upregulated or downregulated) and for their upstream or downstream neighbors located on the same or opposite strand. For each group of genes, the percentage of genes forming gene loops is indicated. *p* values from Fisher exact test were adjusted by the BH method.

Figure 7a. Experimental set-up of the photomorphogenesis experiment. The corresponding RNA-seq data are available in GEO series GSE112442.

Figure 7b. Principal component analysis was performed on the top 500 genes with the highest variance across samples using the DESeq2 plotPCA function[56].

Figure 7c. Intersections of the genes significantly regulated (DESeq2, FDR < 5%) in wild type and in the *bdr1,2,3* triple mutant after 2 or 4 h light, compared to dark, were plotted as Venn diagrams using the Vennerable R package[68].

Figure 7d. RNA-seq data GSE112442 were analyzed with DESeq2[56] to identify differential expression induced by 2 or 4 h light compared to dark in wild type and the *bdr1,2,3* mutant.

From all tandem gene pairs with an intergenic distance <600 bp, we selected those with an upstream gene that was upregulated (fold-change >2, BH-adjusted *p* value <0.05, red, *n* = 278) or not differentially expressed (fold-change <1.5, BH-adjusted *p* value >0.2, blue, *n* = 251) at both time points (2 or 4 h) and in both wild type and the *bdr1,2,3* triple mutant. We also removed genes showing evidence (*p* < 0.05) of a pre-existing TI (downregulation in *bdr1,2,3* vs wt) under the dark condition because increased TI might be hard to detect for these genes (final number of genes, *n* = 214 controls and *n* = 231 genes with an upregulated upstream tandem neighbor). Then we plotted as boxplots the distribution of log(*bdr1,2,3*/wt) values in each condition (dark, light 2 h, and light 4 h) for both groups of genes.

Figure 7e. From all tandem gene pairs with an intergenic distance <600 bp, we selected those with a downstream gene showing some evidence (*p* < 0.05) of TI (i.e., downregulated in *bdr1,2,3* vs wt) under the dark condition and an upstream gene that was either downregulated (fold-change >1.5, BH-adjusted *p* value < 0.05, green, *n* = 148) in both wild-type and the *bdr1,2,3* triple mutant at 2 or 4 h or were not differentially expressed (fold-change <1.5, BH-adjusted *p* value >0.1, blue, *n* = 70) in any genotype and at any time points. Then we plotted as boxplots the distribution of log(*bdr1,2,3*/wt) values in each condition (dark, light 2 h, and light 4 h) for both groups of genes.

We used Wilcoxon signed-rank test to evaluate the change in log2(*bdr1,2,3*/wt) between the dark and light conditions. All *p* values <0.05 are reported.

Figure 7f. Using RNA-seq data GSE112442, we selected all genes that were upregulated by light at 2 or 4 h in wild-type plants (DESeq2; fold-change >1.5, BH-adjusted *p* value <0.05) and separated them by both orientation of their upstream gene (O: upstream gene is on the opposite strand, and S: upstream gene is on the same strand) and intergenic distance between their TSS and the upstream gene border (<600 bp; between 600 and 1200 bp or >1200 bp). For each group of genes and under each condition (dark, light 2 h, and light 4 h), we plotted the distribution of the log2(*bdr1,2,3*/wt) for the downstream genes and compared genes with an upstream neighbor on the same strand (S) to those having an upstream neighbor on the opposite strand (O) with Mann–Whitney U test with a BH *p* value adjustment. All adjusted *p* values <0.05 are shown.

Figure 7g. The Gviz Bioconductor package[67] was used to plot the coverage tracks for BDR1::MYC and BDR2::MYC (FP10M, GSE113059), Pol II ChIP-seq (FP10M, GSE113078), and RNA-seq data (average RPM from triplicates in each group, GSE112442) for PCK1 (AT3G12780), GAPDH B subunit (AT1G42970), and FBA7 (AT4G26520) and their respective upstream tandem gene neighbor.

**Reporting summary.** Further information on research design is available in the Nature Research Reporting Summary linked to this article.

## Data availability
Raw and processed ChIP-seq and RNA-seq data, along with detailed experimental and bioinformatic procedures, are provided in GEO Series GSE112443 and its subseries. TAIR10 annotations were used for all analyses. All other relevant data supporting the key findings of this study are available within the article and its Supplementary Information files or from the corresponding author upon reasonable request. A reporting summary for this article is available as a Supplementary Information file.

## Code availability
Author-generated computer codes and algorithms are available upon request.

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

## Acknowledgements

Next-generation sequencing was performed at the Indiana University Center for Genomics and Bioinformatics. We thank Mathilde Gorieu for technical help during the photomorphogenesis experiment. We are grateful to the genotoul bioinformatics platform Toulouse Midi-Pyrenees (Bioinfo Genotoul) for providing computing and storage resources. This work was supported by a grant to S.D.M. from the National Institutes of Health (GM075060). P.G.P.M. received the support of the EU in the framework of the Marie-Curie FP7 COFUND People Programme, through the award of an AgreenSkills+ fellowship (under grant agreement no. 609398).

## Author contributions

All authors participated in the design of experiments. X.Y. created genetic materials, X.Y. and P.G.P.M. created sequencing libraries, and P.G.P.M. performed bioinformatic analyses.

## Additional information

**Competing interests:** The authors declare no competing interests.

