## [Peer Review File · Nature Communications]

Reviewers' comments:

Reviewer #1 (Remarks to the Author):

Review for:

BORDER proteins promote 3' Pol II pausing and protect the genome from transcriptional interference

In this work, Yu et al., start by investigating the function of three related Arabidopsis BRD proteins that show homology to the yeast transcriptional regulator protein, Bye1. They show that the BRD1/2/3 proteins function redundantly, and that BRD1 and BRD2 proteins localise to gene flanking regions, with BRD1 in showing a particular enrichment at the TES over the TSS. Pol II ChIP-seq and RNA-seq experiments show that genes requiring BRD1/2/3 for expression are typically located downstream of a gene, on the same strand, that has BRD1/2/3-dependent TES Pol II pausing. The authors suggests a model whereby BRD1/2/3 proteins are require to prevent transcriptional interference from 'terminating' Pol II invading into the promoter region of the next gene. To test this model, they used a light a shift experiment to gauge the effect of manipulating the expression of upstream genes in WT vs *brd1/2/3* backgrounds, again finding evidence that BRD1/2/3 are required to prevent transcriptional interference from an upstream neighbour.

This is an interesting piece of work and the findings described in this research would be of novel interest to the field. The idea that compact genomes such as Arabidopsis would have mechanisms to protect downstream genes from TI is compelling and underappreciated. The major conclusions are generally supported by the data presented. However, there are some sections that are difficult to follow and require further clarification. Comments on the manuscript are provided below:

Detailed comments:

What are the molecular signatures for TI? Could the authors for instance look at small RNA accumulation/5' capping at BRD protected genes in WT vs *brd1/2/3*? This would help to provide direct support for their proposed BRD mechanism of action.

"Shifting in this situation may be facilitated by a relatively nucleosome-free region." This is an interesting idea that would provide insight into potential recruitment mechanisms for BRD proteins, and it would be nice to try to confirm this experimentally, for instance by ATAC-seq or MNase-seq.

How many BRD1/2/3 related proteins are there in Arabidopsis, and what is the phylogenetic relationship? I.e. are BRD1 and 2 most closely related followed by 3?

Why do the authors only look at BRD1 and BRD2 transgenic lines, given that it is the triple mutant that has a root growth phenotype? Do transgenic BRD3 lines show a similar localisation pattern? And are they able to complement? While this is not essential to the story, if the authors know the answer it would be useful to include.

In Fig 2B, what is the mini peak in BRD2 just upstream of the TES? Is it real or a bioinformatic artefact?

“>75% of 21,334 BDR1 peaks were located in or within 50bp of an intergenic region” – is this a statically significant enrichment? In figure 2C it would be good to show the comparison for overlaps with a set of randomly shuffled control peaks, or simply the genome average for these features.

“with 85 % of the ~12,000 BDR2 peaks overlapping BDR1 peaks” Again, it would be good to include some sort of test for significance as this is a large number of peaks and could cover a significant fraction of the genome.

What is the significance/implication of the findings shown Fig 2F? Does it suggest that BRD proteins are targeted to nucleosome free regions of the genome that are enriched in regulatory function? It would be good to explain what is meant by ‘conservation’ i.e. as compared to other species/ecotypes, or in terms of within-genome nucleotide diversity? Also in Fig. 2F, the x-axis label should probably just be ‘distance to peak summit’? Please provide a description in the methods of what the phastCons score is.

Fig 3D. This is a difficult plot to understand. Why in the upper panel is it 100% coverage at the gene of interest, while for the lower panel it is 0% coverage?

What is the rationale for the GO terms shown in Fig 3F? Were these the only significant GO terms categories?

In Fig. 4A, I am not convinced that transcriptionally stable non-DE genes is the best control set here. Perhaps a better control would be a transcriptionally matched set of genes, particularly given that the authors have already shown that the expression level of genes upstream of BRD protected genes tends to be relatively high (Fig 3F - and thus you would expect a distinct Pol II signal) and that 3' pausing correlates with expression (Fig. 3C).

In Fig 4C, is this the profile for all genes or just for BRD protected genes?

In general Fig 5D is quite confusing and needs more explanation/labelling. The right panels are not the reciprocal analysis of the left panels, as the authors have restricted their analysis to genes that have 'TI potential', which basically means that they go down in brd1/2/3 – which is why the genes have a log2 below zero. Whereas in the left panel, they only consider whether the upstream gene goes up or is not DE.

Why not do the reciprocal analysis? Generally it would be good to make it more clear to the reader how the analysis is being conducted.

“It is interesting to note that, although 3' pausing is reduced at upstream genes in bdr1,2,3, the expression level, cleavage, and polyadenylation of the upstream genes themselves generally appear similar to wild type.” – where is the evidence for this? Expression level I guess is from RNAseq (although there is no plot to directly show this?) but where do they look at cleavage or poly-A in the manuscript? If they do not have evidence for this I suggest they remove this line from the manuscript.

Fig S2. In the middle and right panels, are the x-axis labels for TSS and TES correct or is it as indicated by the arrow?

Reviewer #2 (Remarks to the Author):

In this article Yu et al explore the role of Bdr proteins in gene expression. They find that these proteins localize preferentially at intergenic regions and their integrity is important for RNAPII pausing at the 3' end of genes. They propose that 3' end pausing of RNAPII is necessary to prevent

transcriptional interference events between tandem genes that could affect negatively gene expression. Most of the experiments seem to be carefully performed and support their model. However, this is the shortest paper I have ever read and I have the feeling that 80% of the manuscript is missing. I believe the manuscript should be entirely rewritten to include:

-An introduction describing the biological context and providing the information required to understand this study. For instance, what do we know about Bdr proteins? Are there orthologues in other organisms? Which functions have been assigned to them? What do we know about transcription termination and transcriptional interference in plants?

-A complete description of their results (some results are present in the figures but not mentioned in the text...)

-A discussion of the results that allow readers to understand what is really novel from their data, what is different or similar to other organisms, maybe include some mechanistic speculation, etc.

Some additional comments that might further help improving this manuscript:

-The entire manuscript is modelled from the beginning on the hypothesis that Bdr proteins are negative regulators of transcriptional elongation but according to the text the origin of this idea seems to be simply the presence of a TFIIIS-like domain in Bdr proteins. By the way, TFIIIS is a positive regulator of elongation (not mentioned in the text). Their model might be true, and a good amount of data go in this direction, but readers should arrive to this conclusion after an unbiased and careful analysis of all the results including alternative interpretations for their data. One have the feeling that only the results that fit the model are mentioned.

-Figure 1: I do not work with the Arabidopsis model but I guess the short root phenotype can be the result of many different mutations aside from Bdr mutations. The partial suppression they observe in the presence of MPA and 6AU, because of the dramatic impact of these compounds on gene expression, may support the authors' favorite hypothesis but also several alternative ones.

-It is unclear whether the authors know or suppose that Bdr proteins associate with RNAPII. If this is not known, a simple coimmunoprecipitation experiment could clarify this important aspect and provide some more mechanistic details to this study that, although interesting, remains quite descriptive. If Bdr proteins interact with RNAPII, it would be convenient to have a look to the Bdr CHIP data side by side with RNAPII CHIP and also to normalize the Bdr data on RNAPII CHIP.

-Main text, related to figure 2: "...intergenic sequence conservation was significantly higher at BDR1 peaks compared to surrounding intergenic sequences, other nucleosome free regions, or random intergenic sequences." I do not understand what this means.

-Figure 3: it is unclear why the authors map specifically the S5P and the S2P form of RNAPII by ChIP and according to the text these experiments do not seem to provide any additional information relative to the results with total RNAPII.

-Figure 3D-F: what are those BDR-repressed genes? Why there is no comment on those genes in the text? How the authors interpret this repression?

-Figure 4: not only 3' end pausing but also 5' end pausing seem more prominent in genes immediately upstream of BDR-protected genes but the authors completely ignore this.

-In the organisms where transcriptional interference has been studied in more detail, it has been shown that it is not directly the presence of the RNAPII at the promoter region of a downstream promoter what induces repression, but rather the accumulation of repressive chromatin marks (as a consequence of transcription) or eventually the accumulation of nucleosomes at the downstream promoter, which can preclude the recognition of binding sequences by transcriptional activators. It would be important to understand if in the absence of Bdr proteins chromatin changes at the downstream promoter can be detected.

- Last paragraph: "...although 3' pausing is reduced at upstream genes in *bdr1,2,3*, the expression level, cleavage, and polyadenylation of the upstream genes themselves generally appear similar to wild type". I do not see any experiment or analysis in the manuscript that allow the authors to conclude this.

Reviewer #3 (Remarks to the Author):

The main findings reported are that in Arabidopsis, so-called Border proteins locate to intergenic regions to protect the expression of downstream genes from transcription interference from PolII transcribed upstream genes.

The data shown is mostly convincing and the manuscript is very well written. This definition of a fundamental feature of plant biology reinforcing genome organization is welcome. The relative weakness of the study would appear to be the limited impact of loss of Border protein function. Although the triple mutants have a clear short-root phenotype, the impact of the mutations appear modest on individual genes in the data as presented.

I have some minor suggestions that might improve the manuscript.

1. I'd like to see the relatedness of the Border proteins to previously characterized negative elongation factors (as introduced in paragraph 2) illustrated with a figure on the related nature of the domain organization of the proteins from different species.
2. In Figure 1 please show both parts of the error bars (ie not dynamite plots)
3. In the ChIP-Seq analysis - Figure 2D the number of input genes is different to Figure 2E – why is this?
4. Greater clarity in the diversity of species used for the phastCons score plot in Figure 2F would be welcome (rather than being directed to a reference). It is notable that such a peak of conservation would be found in intergenic space – does no motif underlie this conservation? Can the authors comment on this.
5. To what extent are the metagene plots in Figure 3a an artefact of the sliding window analysis? A small peak is seen at the TES for S2P (where one would expect to find it) as well as S2P (where one wouldn't expect to find it). The derived Figure 2c is then overly complex and the key finding here relating to pausing is difficult to interpret.
6. In general the illustrative gene examples eg Fig 4D are not that compelling. A Log scale might help to clarify differences in the RNA-Seq data, for example, but the differences in the PolII ChIP data are very modest. Can the authors comment on the PolII ChIP data presented at individual genes
7. A weakness of the manuscript is the apparent relative lack of impact of the border proteins mutations. The final section of the manuscript takes the shift of seedlings from dark-to light growth as an experimental system, because it is accompanied by huge changes in patterns of gene expression. However, the take home message of this section is that this gene expression programme proceeds essentially normally in the absence of the border proteins (see5C) and where changes in

gene expression are detected, the impact appears modest on a small number of genes (5D and E) and major differences in the PolIII ChIP in individual cases is unclear (5F).

8. Mutation of Arabidopsis 3' end processing factors such as conserved poly(A) factors, or FCA and FPA has been shown to be associated with downstream readthrough and chimeric RNA formation. This suggests that Border proteins are not sufficient to limit elongation into downstream genes in these circumstances. Is this the case, or do Border proteins not locate to the regions sensitive to read-through in these mutant backgrounds?

9. The authors conclude that cleavage and polyadenylation of upstream genes appears normal, but do not clarify how this was determined.

10. How do the authors propose that Border proteins function to control PolIII invasion of downstream genes?

11. How do the authors propose Border proteins are recruited to short intergenic regions?

RESPONSE TO REVIEWERS

We would like to sincerely thank the reviewers for their careful and thorough examination of our manuscript. Nearly all of their suggestions have been incorporated into our revised manuscript. We have made significant revisions throughout the manuscript including, the addition of BDR3 ChIP-seq data, 7 additional figures (2 in the main body and 5 supplemental), a longer introduction, subheadings in Results, a Discussion, and expanded figure legends. We hope that these changes will address the reviewer's concerns. Author responses to specific reviewer comments are shown below in BLUE.

Reviewer #1 (Remarks to the Author):

Review for:

BORDER proteins promote 3' Pol II pausing and protect the genome from transcriptional interference

In this work, Yu et al., start by investigating the function of three related Arabidopsis BRD proteins that show homology to the yeast transcriptional regulator protein, Bye1. They show that the BRD1/2/3 proteins function redundantly, and that BRD1 and BRD2 proteins localise to gene flanking regions, with BRD1 in showing a particular enrichment at the TES over the TSS. Pol II ChIP-seq and RNA-seq experiments show that genes requiring BRD1/2/3 for expression are typically located downstream of a gene, on the same strand, that has BRD1/2/3-dependent TES Pol II pausing. The authors suggests a model whereby BRD1/2/3 proteins are require to prevent transcriptional interference from 'terminating' Pol II invading into the promoter region of the next gene. To test this model, they used a light a shift experiment to gauge the effect of manipulating the expression of upstream genes in WT vs *brd1/2/3* backgrounds, again finding evidence that BRD1/2/3 are required to prevent transcriptional interference from an upstream neighbour.

This is an interesting piece of work and the findings described in this research would be of novel interest to the field. The idea that compact genomes such as Arabidopsis would have mechanisms to protect downstream genes from TI is compelling and underappreciated. The major conclusions are generally supported by the data presented. However, there are some sections that are difficult to follow and require further clarification. Comments on the manuscript are provided below:

Detailed comments:

R1.1. What are the molecular signatures for TI? Could the authors for instance look at small RNA accumulation/5' capping at BRD protected genes in WT vs *brd1/2/3*? This would help to provide direct support for their proposed BRD mechanism of action.

Because the term transcriptional interference is applied rather broadly to describe a range of phenomena in which one gene negatively affects the expression of a nearby gene, there really aren't well defined molecular signatures. In our specific instance, the BDR-protected genes have a fairly clear molecular phenotype. The upstream neighbors of BDR-protected genes have high levels of 3' pausing. In the *brd1,2,3* mutant, 3' pausing is reduced and peak Pol II occupancy shifts ~100bp downstream into the promoter region of the BDR-protected gene. We speculate that this "invasion" of Pol II into the promoter region of the downstream gene prevents the recruitment and/or assembly of transcription machinery at

the downstream gene. In this model, we predict that transcription is not initiated at the downstream gene. Therefore, we would not expect to find the accumulation of small 5' capped RNAs from BDR-protected gene (which would suggest that transcription is initiated at BDR-protected genes, but for some reason not completed). To improve our manuscript, we have expanded our discussion of TI and our model. Also, in response to a comment from Reviewer 3, added some data that helps to further clarify the phenotype of *bdr1,2,3*. Mutations in the RNA-binding protein FPA lead to readthrough transcription of neighboring genes and the production of chimeric transcripts. We looked for evidence of chimeric transcripts in *bdr1,2,3* mutants, but found none (2 loci in Fig. 6D, 9 loci in Fig. S7). We added browser shots of an additional 9 BDR-protected genes to illustrate this point.

R1.2. “Shifting in this situation may be facilitated by a relatively nucleosome-free region.” This is an interesting idea that would provide insight into potential recruitment mechanisms for BRD proteins, and it would be nice to try to confirm this experimentally, for instance by ATAC-seq or MNase-seq.

We attempted to explore this possibility by plotting signals from H3 ChIP-seq and MNase-seq in the regions downstream of the TES of the upstream tandem neighbors of BDR-protected genes or control genes (below). Unfortunately for our model, we do not see evidence of decreased nucleosome density downstream of the upstream tandem neighbors of BDR-protected genes. In fact, nucleosome density increases more rapidly downstream of the upstream tandem neighbors of BDR-protected genes. This is likely due to shorter intergenic distances between BDR-protected genes and their upstream neighbors and the well positioned +1 nucleosomes of the BDR-protected genes. Because these data do not support our model, this speculation has been removed from the manuscript.

R1.3. How many BRD1/2/3 related proteins are there in Arabidopsis, and what is the phylogenetic relationship? I.e. are BRD1 and 2 most closely related followed by 3?

We have expanded the text to make it clearer that there are three BDR proteins and added a sequence alignment (Fig. S1). BDR1 and BDR2 are more closely related to each other than to BDR3. We have also added comparisons to similar proteins found in humans and yeast (Fig. 1A).

R1.4. Why do the authors only look at BRD1 and BRD2 transgenic lines, given that it is the triple mutant that has a root growth phenotype? Do transgenic BRD3 lines show a similar localisation pattern? And are they able to complement? While this is not essential to the story, if the authors know the answer it would be useful to include.

Due to an error in the genome annotation, our initial BDR3 constructs were not functional. We have corrected the error and now have BDR3 constructs that rescue the *bdr1,2,3* phenotype and have been able to add BDR3 ChIP-seq data to the manuscript (Figs. 2, 3, 6, S2, S3, and S6). Interestingly, unlike BDR1 and BDR2 which bind strongly near both the TSS and TES, BDR3 shows relatively little binding near the TSS (Fig. 2B).

R1.5. In Fig 2B, what is the mini peak in BRD2 just upstream of the TES? Is it real or a bioinformatic artefact?

This may have been a bioinformatic artifact. The version of this figure in the original submission used un-normalized BDR2 coverage. When we normalized the BDR2::MYC coverage to the wild-type non-transgenic control, the minipeak was no longer visible. As a result, we have normalized all BDR ChIP-seq data in the revised manuscript.

R1.6. “>75% of 21,334 BDR1 peaks were located in or within 50bp of an intergenic region” – is this a statically significant enrichment? In figure 2C it would be good to show the comparison for overlaps with a set of randomly shuffled control peaks, or simply the genome average for these features.

We changed the way this data is presented to make it more clear (Fig. 2D). As suggested, we have added the distribution of each genomic feature in the whole genome, and have calculated the statistical significance of the enrichment for each genomic feature using 10,000 random samples of genomic positions.

R1.7. “with 85 % of the ~12,000 BDR2 peaks overlapping BDR1 peaks” Again, it would be good to include some sort of test for significance as this is a large number of peaks and could cover a significant fraction of the genome.

Due to the addition of the BDR3 ChIP-seq data, we have changed the way the data is presented. We now focus on the differences in overlap between the three BDR proteins. We have added Venn diagrams to compare the overlap between all BDR1, BDR2, and BDR3 peaks (Fig. 2C) or specifically for those peaks occurring near the TSS (Fig. 2E) or TES (Fig. 2F). Consistent with their greater amino acid similarity, greater overlap is observed between BDR1 and BDR2.

R1.8. What is the significance/implication of the findings shown Fig 2F? Does it suggest that BRD proteins are targeted to nucleosome free regions of the genome that are

enriched in regulatory function? It would be good to explain what is meant by 'conservation' ie as compared to other species/ecotypes, or in terms of within-genome nucleotide diversity? Also in Fig. 2F, the x-axis label should probably just be 'distance to peak summit'? Please provide a description in the methods of what the phastCons score is.

We have expanded the discussion of the phastCons analysis (now Fig. 3B,C,D) to include more explanation of the method. We have also added an analysis of conserved motifs in the conserved regions. We have corrected the x-axis label as well (thanks to the reviewer for catching it).

R1.9. Fig 3D. This is a difficult plot to understand. Why in the upper panel is it 100% coverage at the gene of interest, while for the lower panel it is 0% coverage?

The 100% coverage vs 0% coverage was due to the details of how the analysis software dealt with genes on the forward and reverse strands. We agree that it was unnecessarily confusing. The figure (now 5A) has been edited to make the two panels consistent and easier to understand.

R1.10. What is the rationale for the GO terms shown in Fig 3F? Were these the only significant GO terms categories?

Correct, these were the only significantly enriched GO categories. We have added text to the figure legend (now 5C) to more clearly indicate that "Categories with $p < 0.001$ are shown."

R1.11. In Fig. 4A, I am not convinced that transcriptionally stable non-DE genes is the best control set here. Perhaps a better control would be a transcriptionally matched set of genes, particularly given that the authors have already shown that the expression level of genes upstream of BRD protected genes tends to be relatively high (Fig 3F - and thus you would expect a distinct Pol II signal) and that 3' pausing correlates with expression (Fig. 3C).

This is an excellent point. We identified a set of 1,500 control genes with an expression distribution that matches the upstream neighbors of BDR-protected genes (Fig. S5). This new set of "expression-matched" controls has been added (Fig. 6A, 6B, S6). In general, the results are similar to the original non-DE controls. The magnitude of the differences is somewhat attenuated, particularly near the TSS, but the higher enrichment of Pol II and BDR proteins near the TES of the tandem upstream neighbors of BDR-protected genes is still apparent (Fig. 6A,B).

R1.12. In Fig 4C, is this the profile for all genes or just for BRD protected genes?

It is only for BDR-protected genes. We have expanded the legend (now Fig. 6C) to make this more clear:

"C) 3' paused Pol II at upstream genes is reduced and shifted downstream in the absence of BDR proteins. Pol II near the TES of tandem upstream neighbors of BDR-protected genes

is shifted ~96bp downstream in the *bdr1,2,3* mutant. Average Pol II ChIP-seq profiles are presented.”

R1.13. In general Fig 5D is quite confusing and needs more explanation/labelling. The right panels are not the reciprocal analysis of the left panels, as the authors have restricted their analysis to genes that have ‘TI potential’, which basically means that they go down in *brd1/2/3* – which is why the genes have a log2 below zero. Whereas in the left panel, they only consider whether the upstream gene goes up or is not DE. Why not do the reciprocal analysis? Generally it would be good to make it more clear to the reader how the analysis is being conducted.

The reviewer is correct that the left and right portions of 5D are not reciprocal experiments. As the reviewer points out, this can lead to confusion. Therefore, we have split these data into separate panels (now Fig. 7D,E) with separate, and significantly expanded, figure legends (below). The logic of the experiments is hopefully more clear now. In 7D, we are looking for “new” cases of TI that occur in *bdr1,2,3* when the upstream gene is activated by light. This is why we removed genes which already showed evidence of a pre-existing transcriptional interference (downregulation in *bdr1,2,3* vs wt) in darkness. In 7E, we are looking for evidence that pre-existing TI (i.e., observed in darkness) can be attenuated when the upstream gene is down regulated by light.

“D) BDR proteins help prevent TI when nearby tandem upstream genes are upregulated by light. From all tandem gene pairs with an intergenic distance below 600bp, we selected those with upstream genes that were either upregulated or not differentially expressed at both 2h or 4h light. Boxplots show decreased relative expression ($\log[bdr1,2,3/wt]$ values) of the downstream gene when the upstream gene is upregulated by light. Significant changes of $\log_2(bdr1,2,3/wt)$ are evaluated by Wilcoxon signed rank test.

E) Downregulation of the upstream tandem neighbor by light can attenuate TI in *bdr1,2,3*. From all tandem gene pairs with an intergenic distance below 600bp, we selected those with a downstream gene showing some evidence ($p < 0.05$) of transcriptional interference (i.e. downregulated in *bdr1,2,3* vs wt) under the darkness. Tandem pairs were then selected where the upstream gene showed either downregulation or no change in expression at either 2h or 4h light. Boxplots show the upregulation of the downstream gene when the upstream gene is repressed by light ($\log[bdr1,2,3/wt]$ values). Significant changes of $\log_2(bdr1,2,3/wt)$ are evaluated by Wilcoxon signed rank test.”

R1.14. “It is interesting to note that, although 3’ pausing is reduced at upstream genes in *bdr1,2,3*, the expression level, cleavage, and polyadenylation of the upstream genes themselves generally appear similar to wild type.” – where is the evidence for this? Expression level I guess is from RNAseq (although there is no plot to directly show this?) but where do they look at cleavage or poly-A in the manuscript? If they do not have evidence for this I suggest they remove this line from the manuscript.

Because this statement was based largely on visual examination of browser tracks and not a thorough bioinformatic analysis, we agree with the reviewer and have remove it from the manuscript.

R1.15. Fig S2. In the middle and right panels, are the x-axis labels for TSS and TES correct or is it as indicated by the arrow?

We are grateful for the reviewer's keen eye. The axis labels have been corrected (now Fig. S6).

Reviewer #2 (Remarks to the Author):

In this article Yu et al explore the role of Bdr proteins in gene expression. They find that these proteins localize preferentially at intergenic regions and their integrity is important for RNAPII pausing at the 3' end of genes. They propose that 3' end pausing of RNAPII is necessary to prevent transcriptional interference events between tandem genes that could affect negatively gene expression. Most of the experiments seem to be carefully performed and support their model. However, this is the shortest paper I have ever read and I have the feeling that 80% of the manuscript is missing. I believe the manuscript should be entirely rewritten to include:

R2.1. An introduction describing the biological context and providing the information required to understand this study. For instance, what do we know about Bdr proteins? Are there orthologues in other organisms? Which functions have been assigned to them? What do we know about transcription termination and transcriptional interference in plants? A complete description of their results (some results are present in the figures but not mentioned in the text...). A discussion of the results that allow readers to understand what is really novel from their data, what is different or similar to other organisms, maybe include some mechanistic speculation, etc.

For initial review, Nature Communications does not require reformatting of manuscripts that have been considered for publication elsewhere, which was the case for our manuscript. In our revised version, we have taken advantage Nature Communications' more generous page limits. Our manuscript is more than double the length of the original, with 7 additional figures (2 in the main body and 5 supplemental). We have also added a longer introduction, subheadings throughout the results, a discussion, and significantly expanded figure legends. We hope that these changes have addressed the reviewer's concerns.

Some additional comments that might further help improving this manuscript:

R2.2. The entire manuscript is modelled from the beginning on the hypothesis that Bdr proteins are negative regulators of transcriptional elongation but according to the text the origin of this idea seems to be simply the presence of a TFIIIS-like domain in Bdr proteins. By the way, TFIIIS is a positive regulator of elongation (not mentioned in the text). Their model might be true, and a good amount of data go in this direction, but readers should arrive to this conclusion after an unbiased and careful analysis of all the results including alternative interpretations for their data. One have the feeling that only the results that fit the model are mentioned.

We have expanded the introduction and results to include a discussion of TFIIIS and a longer discussion of the relationships of the BDR proteins to TFIIIS and Bye1 (negative elongation factor). We hope this will give the reader a better introduction to BDR-related proteins:

“BDR proteins form a three-member family in Arabidopsis (BDR1=At5g25520, BDR2=At5g11430, BDR3=At2g25640). Each BDR protein contains a SPOC domain, which is found the SPEN family of transcriptional repressors, and a Transcription Elongation Factor IIS (TFIIS) central domain (Figs. 1A, S1)^{12,13}. TFIIS contains three domains (I, II/central, and III) and acts as a positive elongation factor. During elongation, RNA Pol II frequently backtracks, such that it is no longer positioned at the 3' end of the growing transcript. To restart elongation, the central domain of TFIIS binds to RNA Pol II, while domain III stimulates cleavage of the nascent transcript, thus providing a new 3' end for RNA Pol II^{12,14-16}. The fact that BDR proteins do not contain domain I or III, suggests that the BDR proteins are unlikely to have TFIIS-like activity.

Proteins with similar domain organization are found outside plants, with fungal and animal proteins often including an additional N-terminal PHD domain¹⁷⁻¹⁹ (Fig. 1A). These include the mammalian proteins SPOCD1, PHF3, and DIDO1^{17,20,21}. The best characterized is the yeast protein BYpass of Ess1 (Bye1), which contains a PHD domain in addition to its SPOC and TFIIS central domains. Bye1 is thought to act as a negative elongation factor and binds to Pol II through its TFIIS central domain and to histone H3 trimethylated on lysine 4 (H3K4me3) through its PHD domain^{17,22}. Bye1 is enriched in the 5' regions of genes^{17,23} and, consistent with a role in repressing Pol II elongation, Pol II occupancy in the 5' regions of genes is reduced in the *bye1* mutant, whereas Pol II occupancy is increased in gene bodies²².

R2.3. Figure 1: I do not work with the Arabidopsis model but I guess the short root phenotype can be the result of many different mutations aside from Bdr mutations. The partial suppression they observe in the presence of MPA and 6AU, because of the dramatic impact of these compounds on gene expression, may support the authors' favorite hypothesis but also several alternative ones.

It is certainly the case that other mutations can cause reduced root growth in Arabidopsis. We believe the significant result is that two different chemicals used as inhibitors of transcriptional elongation cause shorter roots in wild type, but longer roots in *bdr1,2,3*. I.e., the chemicals have opposite effects on root growth in wild type vs *bdr1,2,3*. There may be other explanations for our results, of course, but no specific alternative mechanisms come to mind that would explain reduced growth in wild type and increased growth in the mutant. We've expanded the discussion of these experiments and our conclusion is only that our data “suggests” that the short-root phenotype may be due to increased transcriptional elongation in the *bdr1,2,3* background.

R2.4. It is unclear whether the authors know or suppose that Bdr proteins associate with RNAPII. If this is not known, a simple coimmunoprecipitation experiment could clarify this important aspect and provide some more mechanistic details to this study that, although interesting, remains quite descriptive. If Bdr proteins interact with RNAPII, it would be convenient to have a look to the Bdr ChIP data side by side with RNAPII ChIP and also to normalize the Bdr data on RNAPII ChIP.

These are good points. We have complete immunoprecipitation/mass spectrometry (IP-MS) experiments with BDR1 and BD2. Both proteins identified peptides from the 3rd-

largest subunit of Pol II; no Pol II proteins were identified in wild-type controls. This data has been added as Supplemental Table 1. We have also included ChIP-seq analyses where BDR signal has been normalized by Pol II (Fig. S6)

R2.4. Main text, related to figure 2: "...intergenic sequence conservation was significantly higher at BDR1 peaks compared to surrounding intergenic sequences, other nucleosome free regions, or random intergenic sequences." I do not understand what this means.

We have expanded the discussion of the phastCons analysis (now Fig. 3B,C,D) and have also added an analysis of conserved motifs in the conserved regions:

"We also examined sequence conservation around BDR peaks. PhastCons²⁶ examines sequence conservation between Arabidopsis and the genomes of 20 other angiosperms. In order to focus on the conservation of intergenic regions, sequences corresponding to annotated genes were removed. Because BDR peaks are preferentially found in nucleosome-depleted regions, we included other nucleosome-free regions, as well as random intergenic sequences, as controls. We found that sequence conservation was significantly higher at BDR peaks compared to surrounding intergenic sequences (Fig. 3B), with higher conservation observed for BDR1 and BDR2 peaks than for BDR3 peaks. We also searched for over-represented motifs in BDR1 and BDR2 peaks, focusing on the 101bp surrounding the peak center. Two motifs were identified that occurred more frequently in BDR1 and BDR2 peaks than in other intergenic regions (Fig. 3C,D). A TCP-like motif was found in 44.9% of BDR1 peaks and an E-box motif was found in 7.02%²⁷. Both motifs were also enriched in BDR2 peaks (Fig. 3D). Interestingly, although BDR1 and BDR2 are enriched near both TSS and TES regions (Fig. 2B,E,F), these motifs only show enrichment near TSS sites (Fig. 3C). Because BDR proteins lack characterized DNA-binding motifs, it is likely that recruitment to chromatin depends on interactions with other factors. The result that enriched sequence motifs are found near at the TSS, but not the TES, suggests that BDR proteins may be recruited to chromatin through multiple interactions/mechanisms. E.g., interacting with DNA-binding proteins that recognize TCP-like and/or E-box motifs near the TSS, and other proteins, such as components of the transcription termination machinery near the TES. The model that BDR proteins may be recruited to TSS and TES regions through separate mechanisms is also supported by asymmetric binding profile of BDR3, which shows much stronger affinity for TES regions than TSS sites (Fig. 2B)."

R2.5. Figure 3: it is unclear why the authors map specifically the S5P and the S2P form of RNAPII by ChIP and according to the text these experiments do not seem to provide any additional information relative to the results with total RNAPII.

We used the S5P and S2P antibodies in hopes that they would provide more detail regarding Pol II occupancy at the TSS and TES respectively. Although the ChIP-seq profiles overlap significantly with the Pol II, they do show some distinct features. We have expanded the text to describe the differences in more detail:

"We determined Pol II occupancy in wild type and *bdr1,2,3* using antibodies recognizing Pol II, Serine 5 phosphorylated Pol II (S5P), and Pol II S2P. During transcription, Pol II undergoes a series of phosphorylation events, with Pol II S5P associated with initiation and Pol II S2P associated with elongation²⁸. Consistent with this model, we observed that Pol II

S2P signal increased through the body of the gene (Fig. 4A). S5P occupancy increased through the body of the gene, but also showed a peak near the TSS and a depletion near the TES (Fig. 4A). Consistent with published ChIP-seq, GRO-seq, and pNET-seq studies^{10,11,29}, all three antibodies showed 3' Pol II accumulation just after the TES (Fig. 4A, red arrows), indicative of 3' pausing.”

We agree with the reviewer that in several of the figures (e.g., 4C, 6A,C,E), individual antibodies do not uniquely contribute to the interpretation of the data (i.e., they all provide similar results). Even in these cases, however, observing similar results with three independent antibodies does contribute to the robustness of the data.

R2.5. Figure 3D-F: what are those BDR-repressed genes? Why there is no comment on those genes in the text? How the authors interpret this repression?

In this manuscript, we have focused on the role of the BDR proteins in the promotion of 3' pausing and the protection of downstream tandem genes. We included the BDR-protected and repressed genes in Supplemental Table 3. We have also added a summary figure for expression changes in *bdr1,2,3* as determined by RNA-seq as Fig. S4A.

R2.6. Figure 4: not only 3' end pausing but also 5' end pausing seem more prominent in genes immediately upstream of BDR-protected genes but the authors completely ignore this.

Reviewer 1 suggested using a different set of control genes for this experiment (now Fig. 6A). Because the upstream neighbors of BDR-protected genes have higher-than-average expression, the reviewer suggested using control genes with similar levels of expression (see reviewer comment R1.11). After changing to these “expression-matched” control genes, the differences in 5' Pol II occupancy have largely been eliminated (Fig. 6A). Therefore, the differences seen in the original version of this figure were probably due to the higher expression of the upstream neighbors of BDR-protected genes compared to the non-differentially expressed controls that were originally used.

R2.7. In the organisms where transcriptional interference has been studied in more detail, it has been shown that it is not directly the presence of the RNAPII at the promoter region of a downstream promoter what induces repression, but rather the accumulation of repressive chromatin marks (as a consequence of transcription) or eventually the accumulation of nucleosomes at the downstream promoter, which can preclude the recognition of binding sequences by transcriptional activators. It would be important to understand if in the absence of Bdr proteins chromatin changes at the downstream promoter can be detected.

We appreciate the suggestion. To look at this possibility, we examined H3 ChIP-seq and MNase-seq data centered on the TSS of BDR-protected genes, BDR-repressed genes, expression-matched control genes, and non-differentially expressed control genes (below). Overall, the changes observed between wild type and *bdr1,2,3* were similar across all groups of genes. Thus, in this particular experiment, we did not detect any changes in nucleosome occupancy that correlate with the expression state of the genes.

R2.7. Last paragraph: “...although 3’ pausing is reduced at upstream genes in bdr1,2,3, the expression level, cleavage, and polyadenylation of the upstream genes themselves generally appear similar to wild type”. I do not see any experiment or analysis in the manuscript that allow the authors to conclude this.

Because this statement was based largely on visual examination of browser tracks and not a thorough bioinformatic analysis, we have removed it from the manuscript.

Reviewer #3 (Remarks to the Author):

The main findings reported are that in Arabidopsis, so-called Border proteins locate to intergenic regions to protect the expression of downstream genes from transcription interference from PolII transcribed upstream genes.

The data shown is mostly convincing and the manuscript is very well written. This definition of a fundamental feature of plant biology reinforcing genome organization is welcome. **The relative weakness of the study would appear to be the limited impact of loss of Border protein function. Although the triple mutants have a clear short-root phenotype, the impact of the mutations appear modest on individual genes in the data as presented.**

Addressed in responses to R3.7 and R3.8.

I have some minor suggestions that might improve the manuscript.

R3.1. I’d like to see the relatedness of the Border proteins to previously characterized negative elongation factors (as introduced in paragraph 2) illustrated with a figure on the related nature of the domain organization of the proteins from different species.

As requested, this data has been added to Fig. 1A and S1.

R3.2. In Figure 1 please show both parts of the error bars (ie not dynamite plots)

As requested, error bars have been changed.

R3.3. In the ChIP-Seq analysis - Figure 2D the number of input genes is different to Figure 2E – why is this?

In our original version of these figures, we independently selected all the genes that have both a BDR1 and a BDR2 peak detected within 300bp of their TSS (for 2D) and all the genes that have both a BDR1 and a BDR2 peak within 300bp of their TES (for 2E).

When we added the analysis for BDR3, we decide to alter our strategy for these analyses. Instead of looking for genes with peaks for all three proteins at their TSS or TES, we treated each gene separately. E.g., occupancy of BDR1 over TSS regions containing BDR1 peaks, occupancy of BDR2 over TSS regions containing BDR2 peaks, etc. We believe this provides a more unbiased look at the distribution of BDR proteins.

R3.4. Greater clarity in the diversity of species used for the phastCons score plot in Figure 2F would be welcome (rather than being directed to a reference). It is notable that such a peak of conservation would be found in intergenic space – does no motif underlie this conservation? Can the authors comment on this.

We have expanded the discussion of the phastCons analysis (now Fig. 3B,C,D) and have also added an analysis of conserved motifs in the conserved regions:

“We also examined sequence conservation around BDR peaks. PhastCons²⁶ examines sequence conservation between Arabidopsis and the genomes of 20 other angiosperms. In order to focus on the conservation of intergenic regions, sequences corresponding to annotated genes were removed. Because BDR peaks are preferentially found in nucleosome-depleted regions, we included other nucleosome-free regions, as well as random intergenic sequences, as controls. We found that sequence conservation was significantly higher at BDR peaks compared to surrounding intergenic sequences (Fig. 3B), with higher conservation observed for BDR1 and BDR2 peaks than for BDR3 peaks. We also searched for over-represented motifs in BDR1 and BDR2 peaks, focusing on the 101bp surrounding the peak center. Two motifs were identified that occurred more frequently in BDR1 and BDR2 peaks than in other intergenic regions (Fig. 3C,D). A TCP-like motif was found in 44.9% of BDR1 peaks and an E-box motif was found in 7.02%²⁷. Both motifs were also enriched in BDR2 peaks (Fig. 3D). Interestingly, although BDR1 and BDR2 are enriched near both TSS and TES regions (Fig. 2B,E,F), these motifs only show enrichment near TSS sites (Fig. 3C). Because BDR proteins lack characterized DNA-binding motifs, it is likely that recruitment to chromatin depends on interactions with other factors. The result that enriched sequence motifs are found near at the TSS, but not the TES, suggests that BDR proteins may be recruited to chromatin through multiple interactions/mechanisms. E.g., interacting with DNA-binding proteins that recognize TCP-like and/or E-box motifs near the TSS, and other proteins, such as components of the transcription termination machinery near the TES. The model that BDR proteins may be recruited to TSS and TES regions through separate mechanisms is also supported by asymmetric binding profile of BDR3, which shows much stronger affinity for TES regions than TSS sites (Fig. 2B).”

R3.5. To what extent are the metagene plots in Figure 3a an artefact of the sliding window analysis? A small peak is seen at the TES for S2P (where one would expect to find it) as well as S2P (where one wouldn't expect to find it).

We did not use a sliding window to produce the metagene plots in Fig 3A. On the gene bodies, the coverage is averaged in 100 contiguous bins (no sliding window). In the region before the TSS and after the TSS, we directly plotted the average coverage (and associated CI) at single base resolution (no binning or sliding window used). Thus, we do not believe the peaks are bioinformatic artifacts.

The peak in S5P located near the TES may be in part due to the promoter regions of downstream tandem genes. We have produced the same metagene plots using only genes located >500bp from any other annotated gene. In these plots the TES-associated peak of S5P is strongly attenuated, suggesting that it is due to neighboring genes.

R3.6. The derived Figure 2c is then overly complex and the key finding here relating to pausing is difficult to interpret.

We have presented the data from Figure 2C in a new format (now Fig. 2D), which we hope is easier to understand.

R3.7. In general the illustrative gene examples eg Fig 4D are not that compelling. A Log scale might help to clarify differences in the RNA-Seq data, for example, but the differences in the PolII ChIP data are very modest. Can the authors comment on the PolII ChIP data presented at individual genes

We agree that a problem with our depiction of RNA-seq data in Fig. 4D (now Fig. 6D) is that, because the upstream genes is relatively highly expressed, the differences in expression in the BDR-protected genes are difficult to see clearly. We have re-worked the figure so that each gene is now shown on its own scale. The differences in expression of the BDR-protected gene are now easily seen (Fig. 6D). Regarding the differences in Pol II ChIP-seq signal at individual genes, we have added an additional 9 examples (Fig. S7) and have included both Pol II and S2P tracks. We hope that these additional browser tracks will more clearly show a pattern of reduced 3' Pol II at the upstream tandem neighbors of BDR-protected genes in the *bdr1,2,3* mutant.

R3.8. A weakness of the manuscript is the apparent relative lack of impact of the border proteins mutations. The final section of the manuscript takes the shift of seedlings from dark-to light growth as an experimental system, because it is accompanied by huge changes in patterns of gene expression. However, the take home message of this section is that this gene expression programme proceeds essentially normally in the absence of the border proteins (see5C) and where changes in gene expression are detected, the impact appears modest on a small number of genes (5D and E) and major differences in the PolII ChIP in individual cases is unclear (5F).

All of the points that the reviewer makes are of course true, but not necessarily unexpected. As we show in the manuscript, BDR proteins promote 3' pausing at a large number of genes. Reduced 3' pausing in *bdr1,2,3*, however, only appears to affect gene expression in a specific context : genes that are located a short distance downstream of a highly expressed gene on the same strand. Thus it is not surprising that, in the light experiment, a

relatively small number of genes are differentially expressed and that they preferentially occur in this genomic context. I would only point out that “impact” can sometimes be difficult for scientists to assess in the lab. The short-root phenotype or the slower/reduced upregulation of CBB-cycle genes may seem minor, but even a modest decrease in fitness can lead to an evolutionary dead end.

R3.9. Mutation of Arabidopsis 3' end processing factors such as conserved poly(A) factors, or FCA and FPA has been shown to be associated with downstream readthrough and chimeric RNA formation. This suggests that Border proteins are not sufficient to limit elongation into downstream genes in these circumstances. Is this the case, or do Border proteins not locate to the regions sensitive to read-through in these mutant backgrounds?

We have added a discussion of read-through and the generation of chimeric transcripts in the *fpa* mutant (below). We have visually examined our RNA-seq data at the loci shown to create read-through transcripts in *fpa*, and found no evidence of read-through in *bdr1,2,3*. We also have added browser tracks for an additional 9 examples of BDR-protected genes, which also show no evidence of chimeric RNAs (Fig. S7).

“To investigate the later possibility, we looked for evidence of chimeric readthrough transcripts, which have been reported in mutants for the RNA-binding protein *fpa*³⁰. We saw no evidence, however, of chimeric transcripts between BDR-protected genes and their upstream tandem neighbors (Fig. 6D and S7). This suggests that the reduced expression of downstream tandem genes in *bdr1,2,3* may be due to a failure of transcription factors to assemble at the promoter. Interestingly, while BDR proteins affect the magnitude of 3' pausing for a large fraction of the genome (Fig. 4C), a significant shift in the position of 3' pausing was consistently observed only for the upstream neighbors of BDR-protected genes (Fig. 6E).”

R3.10. The authors conclude that cleavage and polyadenylation of upstream genes appears normal, but do not clarify how this was determined.

Because this statement was based largely on visual examination of browser tracks and not a thorough bioinformatic analysis, we have removed it from the manuscript.

R3.11. How do the authors propose that Border proteins function to control PolIII invasion of downstream genes?

This is an interesting question that we hope to pursue in the future. We have added a bit of speculation to the discussion:

“The precise mechanism by which BDR proteins promote 3' Pol II pausing is unclear; however, it is tempting to speculate that an increased Pol II elongation rate in *bdr1,2,3* mutant might be responsible for a shift in Pol II termination site, as a downstream shift in termination has been observed using a “fast” elongating Pol II in human cells⁴⁵.”

R3.12. How do the authors propose Border proteins are recruited to short intergenic regions?

Two experiments that have been added to the revised manuscript shed some light on how BDR proteins might be recruited to chromatin. 1) We identified sequence motifs that are over-represented in BDR peaks (Fig. 3). Interesting though, although BDR1 and BDR2 bind near both TSS and TES regions, the motifs are only enriched near the TSS. This suggests that BDR proteins may be recruited to the TSS and TES through separate mechanisms. 2) We have added ChIP-seq data for BDR3. Unlike BDR1 and BDR2, BDR3 has a strong preference for TES regions over TSS regions (Fig. 2B). This also supports the model that recruitment to the TSS and TES may occur by independent mechanisms (i.e., it is possible to bind to one region and not the other). The added text is shown here:

“We also searched for over-represented motifs in BDR1 and BDR2 peaks, focusing on the 101bp surrounding the peak center. Two motifs were identified that occurred more frequently in BDR1 and BDR2 peaks than in other intergenic regions (Fig. 3C,D). A TCP-like motif was found in 44.9% of BDR1 peaks and an E-box motif was found in 7.02%²⁷. Both motifs were also enriched in BDR2 peaks (Fig. 3D). Interestingly, although BDR1 and BDR2 are enriched near both TSS and TES regions (Fig. 2B,E,F), these motifs only show enrichment near TSS sites (Fig. 3C). Because BDR proteins lack characterized DNA-binding motifs, it is likely that recruitment to chromatin depends on interactions with other factors. The result that enriched sequence motifs are found near at the TSS, but not the TES, suggests that BDR proteins may be recruited to chromatin through multiple interactions/mechanisms. E.g., interacting with DNA-binding proteins that recognize TCP-like and/or E-box motifs near the TSS, and other proteins, such as components of the transcription termination machinery near the TES. The model that BDR proteins may be recruited to TSS and TES regions through separate mechanisms is also supported by asymmetric binding profile of BDR3, which shows much stronger affinity for TES regions than TSS sites (Fig. 2B).”

REVIEWERS' COMMENTS:

Reviewer #1 (Remarks to the Author):

The authors now include ChIP-seq data from the final family member, BRD3, and have improved and/or clarified a number of their analyses to support their conclusions. Overall, they have satisfactorily addressed my concerns in this revised version of the manuscript.

Reviewer #2 (Remarks to the Author):

In general the new additions to the manuscript provide substantial improvement to the paper and address my main concerns. Just a few comments about points that I believe should be corrected before publication of the manuscript:

Fig S3: In addition to the heatmaps, the authors should provide some statistical analyses on the correlation between the positioning of Bdr proteins and RNAPII and nucleosome-free regions. Although the correlation seems quite clear for Bdr1 (visually), it is not the same for Bdr2 and Bdr3 (actually, in the case of these two proteins, they seem to locate at nucleosomes).

Table S1: the complete list of proteins identified in both the control and the tagged plants should be provided (i.e. not only the values corresponding to the subunits of RNAPII). The scores for each peptide as well as any other relevant information should also be included.

Page 4, lines 23 to 41: A couple of sentences presenting the different groups of genes that are going to be analysed and discussed would be welcome (e.g. "we distinguished three groups of genes: Bdr-protected (lower expression in the Bdr mutants), Bdr-repressed (higher expression in the Bdr mutants) and non-differentially expressed"). Also, it's a bit confusing to define as both Bdr-promoted and Bdr-protected the same group of genes. It's better to choose only one way to call these genes

In its current form, the manuscript contains both a "results and discussion" section and a separate "discussion" section.

Reviewer #3 (Remarks to the Author):

The revised manuscript is greatly improved and our comments have been addressed very well. In response to our comments I'd note two glitches:

1. R3.4: the motif search referred to in the response and in the revised figure legend refers to BRD1 and also BDR2. However, only BDR1 is shown in the actual Figure (3D).
2. R3.7 suggests that PolII and S2P tracks are presented in the revised Figures. However, no S2P track is shown in Fig 6D (it is shown in Supplementary Figures). I think this Figure is improved from the previous version, but I wonder if the authors also considered subtracting the profiles to illustrate the difference in PolII ChIP between wt and the bdr mutants.

Finally, although it wasn't a response to a comment from us, I thought that showing only two PolII peptides (as opposed to the whole IP dataset with statistical analysis) in Table S1 p3 line 31, was a bit weak. It might be best to leave this question open for the moment, if the authors wish to prepare a subsequent publication on the protein interactions of the border proteins.

RESPONSE TO REVIEWER COMMENTS

REVIEWERS' COMMENTS:

Reviewer #1 (Remarks to the Author):

The authors now include CHIP-seq data from the final family member, BRD3, and have improved and/or clarified a number of their analyses to support their conclusions. Overall, they have satisfactorily addressed my concerns in this revised version of the manuscript.

No changes required.

Reviewer #2 (Remarks to the Author):

In general the new additions to the manuscript provide substantial improvement to the paper and address my main concerns. Just a few comments about points that I believe should be corrected before publication of the manuscript:

Fig S3: In addition to the heatmaps, the authors should provide some statistical analyses on the correlation between the positioning of Bdr proteins and RNAPII and nucleosome-free regions. Although the correlation seems quite clear for Bdr1 (visually), it is not the same for Bdr2 and Bdr3 (actually, in the case of these two proteins, they seem to locate at nucleosomes).

This data has been added as Figure S4. We also took the further step of examining the statistical significance of the correlations in 250bp regions immediately before the TSS, after the TSS, before the TES, and after the TES. This analysis helps to show that some correlations are stronger in particular regions. For example, the correlation between BDR3, which shows relatively little binding near the TSS, with BDR1, BDR2, and Pol II is higher near the TES.

Table S1: the complete list of proteins identified in both the control and the tagged plants should be provided (i.e. not only the values corresponding to the subunits of RNAPII). The scores for each peptide as well as any other relevant information should also be included.

This point is addressed under reviewer 3's final comment.

Page 4, lines 23 to 41: A couple of sentences presenting the different groups of genes that are going to be analysed and discussed would be welcome (e.g. "we distinguished three groups of genes: Bdr-protected (lower expression in the Bdr mutants), Bdr-repressed (higher expression in the Bdr mutants) and non-differentially expressed"). Also, it's a bit confusing to define as both Bdr-promoted and Bdr-protected the same group of genes. It's better to choose only one way to call these genes

As requested, we have added a short introduction to the groups of genes used in the analyses:

“We used RNA-seq analysis to identify three sets of genes whose expression is promoted or repressed by BDR proteins (i.e., show decreased or increased expression in *bdr1,2,3* seedlings, respectively), as well as non-differentially expressed genes (Fig. S5A). Interestingly, we found that BDR-promoted genes, which we will refer to as BDR-protected genes, preferentially occur in a specific genomic context (Fig. 5A and S5B).” Following the introduction, we only refer to these as “BDR-protected genes”.

In its current form, the manuscript contains both a “results and discussion” section and a separate “discussion” section.

We have corrected the section headings to “Results” and “Discussion”.

Reviewer #3 (Remarks to the Author):

The revised manuscript is greatly improved and our comments have been addressed very well. In response to our comments I’d note two glitches:

1. R3.4: the motif search referred to in the response and in the revised figure legend refers to BRD1 and also BDR2. However, only BDR1 is shown in the actual Figure (3D).

The data for motif enrichment for BDR2 peaks has been added.

2. R3.7 suggests that PolII and S2P tracks are presented in the revised Figures. However, no S2P track is shown in Fig 6D (it is shown in Supplementary Figures). I think this Figure is improved from the previous version, but I wonder if the authors also considered subtracting the profiles to illustrate the difference in PolII ChIP between wt and the *bdr* mutants.

We have added the S2P track to Fig. 6D.

We have tried subtracting the Pol II ChIP-seq tracks for WT and the *bdr* mutants. The results at a single locus, however, were fairly noisy and did not aid in data interpretation. For comparing the changes in Pol II occupancy between WT and the *bdr* mutants, we feel it is better to use the average signal over groups of genes as we have in 6A-C.

Finally, although it wasn’t a response to a comment from us, I thought that showing only two PolII peptides (as opposed to the whole IP dataset with statistical analysis) in Table S1 p3 line 31, was a bit weak. It might be best to leave this question open for the moment, if the authors wish to prepare a subsequent publication on the protein interactions of the border proteins.

Both reviewer 2 and 3 commented on the BDR-Pol II IP-MS data. This data was added to our revised manuscript in response to a reviewer comment that suggested that we demonstrate an interaction between the BDR proteins, which contain a predicted Pol II interacting domain, and Pol II. Thus, we added this data to address a specific question from one of the three reviewers; it was not our intent to present a complete protein-interaction study. As reviewer 3 speculates, we are preparing a subsequent manuscript that will contain a significant amount of protein-interaction work with the BDR proteins. We agree with reviewer 3 that it is better to present a more thorough analysis in our subsequent publication. We have removed the BDR-Pol II IP-MS data from our manuscript.